# Computing a human-like reaction time metric from stable recurrent vision models

**Lore Goetschalckx**[*1,2], **Lakshmi N. Govindarajan**[*3], **Alekh K. Ashok**[1,2], **Aarit Ahuja**[4],
**David L. Sheinberg**[4], and **Thomas Serre**[1,2]

[1]Department of Cognitive, Linguistic, & Psychological Sciences, Brown University, Providence, RI 02912
[2]Carney Institute for Brain Science, Brown University, Providence, RI 02912
{lore_goetschalckx,alekh_karkada_ashok,thomas_serre}@brown.edu
[3]Integrative Computational Neuroscience (ICoN) Center, MIT, Cambridge, MA 02319
lakshmin@mit.edu
[4]Neuroscience Department, Brown University, Providence, RI 02912
{aarit_ahuja,david_sheinberg}@brown.edu

## Abstract

The meteoric rise in the adoption of deep neural networks as computational models of vision has inspired efforts to "align" these models with humans. One dimension of interest for alignment includes behavioral choices, but moving beyond characterizing choice patterns to capturing temporal aspects of visual decision-making has been challenging. Here, we sketch a general-purpose methodology to construct computational accounts of reaction times from a stimulus-computable, task-optimized model. Specifically, we introduce a novel metric leveraging insights from subjective logic theory summarizing evidence accumulation in recurrent vision models. We demonstrate that our metric aligns with patterns of human reaction times for stimulus manipulations across four disparate visual decision-making tasks spanning perceptual grouping, mental simulation, and scene categorization. This work paves the way for exploring the temporal alignment of model and human visual strategies in the context of various other cognitive tasks toward generating testable hypotheses for neuroscience. Links to the code and data can be found on the project page: https://serre-lab.github.io/rnn_rts_site/.

## 1 Introduction

The symbiosis between visual neuroscience and machine learning has created a distinctive and unprecedented phase in the field of vision. On the one hand, formidable advances in machine learning have supported the development of computational models of visual cognition that rival, and in some cases surpass, human-level performance on naturalistic tasks [1–5]. On the other hand, incorporating computational and neurophysiological constraints from biological visual systems has yielded significant benefits for AI models in terms of their performance [6, 7], efficiency [8], and robustness [9–11]. Additionally, performant computational vision models that operate directly on high-dimensional sensory inputs have shown great promise for scientific discovery through the generation of testable hypotheses for neuroscience [12, 13]. At the core of this synergistic loop lies the need for reliably quantifying the degree of "alignment" between models and the primate visual system, which can subsequently serve as a control knob for closed-loop hypothesis testing.

The notion of "alignment" is subjective and can be operationalized at varying levels of analysis [14]. One of the most prominent alignment techniques is to quantify the degree to which model feature

---

[*]These authors contributed equally to this work.

37th Conference on Neural Information Processing Systems (NeurIPS 2023).

representations predict time-averaged firing rates of single neurons across various stimuli [11, 15–17]. In contrast, computational-level alignment analyses typically use axiomatic attribution methods [18] to extract minimal sub-circuits from larger models and probe them to gain mechanistic insights [12]. A third popular approach to alignment is at a higher behavioral level, where techniques focus on understanding the extent to which models and primates utilize similar "visual strategies". Top-down attention masks, diagnostic of human observers' visual strategies, are a commonly employed axis of comparison in this type of model alignment [7, 19]. At the same time, other behavioral measures such as object confusion patterns [20, 21] or human-derived pairwise image similarity ratings [22] are also widely considered effective quantities that can form the basis of alignment.

Each of these techniques has a common tenet: their purview is limited to measures of alignment that are global, time-independent summaries of neural or behavioral states. While this may be sufficient for task paradigms primarily involving feedforward processing, a large part of visual cognition is highly dynamic [23–25]. Understanding visual computations, thus, necessarily involves characterizing and modeling the time-varying properties of neural responses to complex and varied inputs. *Why, then, is there a dearth of alignment methods focusing directly on the level of visual processing dynamics?* We posit a two-fold bottleneck. First, there is a need for more performant models that exhibit and mimic visual cortical dynamics at scale. Second, we lack the mathematical tools to extract meaningful dynamical signatures from such models and compare them with biological systems. While recent progress in biologically-inspired machine learning has started to open up solutions for the former [8, 19, 26–28], the latter remains a wide-open research area.

**Contributions.** In this study, we consider this temporal alignment perspective. Recent advances in training biologically motivated large-scale convolutional recurrent neural network models (cRNNs henceforth) on visual cognition tasks [8] have provided candidate models to study the temporal alignment problem. A cRNN's inherent design affords access to its time-varying dynamical properties that can be explicitly analyzed. Throughout this paper, we specifically focus on extracting a metric that captures a cRNN's dynamical signature and on quantifying the ability of this metric to explain primate decision time courses in the form of reaction times (RTs) in concomitant visual psychophysics tasks.

- We introduce a novel computational framework to train, analyze, and interpret the behavior of cRNNs on visual cognitive tasks of choice. Our framework triangulates insights from an attractor dynamics-based training routine [8] and evidential learning theory [29] to support stable and expressive evidence accumulation strategies.

- We derive a stimulus-computable, task- and model-agnostic metric to characterize evidence accumulation in cRNNs. Our metric does not require extra supervision and purely leverages internal cRNN activity dynamics.

- We comprehensively demonstrate the efficacy of our metric in capturing stimulus-dependent primate decision time courses in the form of reaction times (RTs) in four disparate visual cognitive challenges that include incremental grouping, mental simulation, and scene categorization. To the best of our knowledge, this is the first demonstration of qualitative temporal alignments between models and primates across task paradigms.

## 2   Related Work

**Cognitive process models.** The speed-accuracy tradeoff in human behavior is ubiquitous in cognitive science [31] and has garnered much attention from the modeling perspective. The most prominent of them is the Drift-Diffusion Model (DDM) family, which specifies RTs by determining when the amount of accumulated evidence for a behavioral choice from a noisy process reaches a threshold value [32, 33]. Though such models have had phenomenal successes in aligning neural and behavioral data [34], they are typically not stimulus-computable or rely on handcrafted heuristics [35]. Moreover, such models are usually directly fitted to RT data [35, 36], unlike our approach, where RT is a derived quantity without any additional supervision.

**Computing reaction-time proxies from neural network models.** Several notable efforts have attempted to derive RT measures from neural network models to serve as a proxy for "computational cost" [26, 37–42]. Spoerer et al. [26] introduced the notion of energy demand in their objective function that motivated parsimony regarding the number of computations the cRNN executes. Bansal

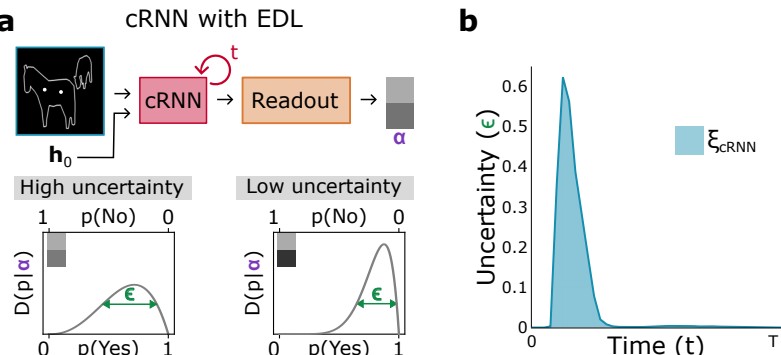

Figure 1: **Computing a reaction time metric from a recurrent vision model. a.** A schematic representation of training a cRNN with evidential deep learning (EDL; [30]). Model outputs are interpreted as parameters ($\alpha$) of a Dirichlet distribution over class probability estimates, with higher values (symbolized here by a darker gray) reflecting more generated evidence in favor of the corresponding class. In this framework, the width of the distribution signals the model's uncertainty ($\epsilon$) about its predictions. **b.** Visualization of our metric, $\xi_{cRNN}$, computed for an example stimulus (see Panel a) used in the task studied in Section 4. The metric, denoted $\xi_{cRNN}$, is defined as the area under the uncertainty curve, i.e., the evolution of uncertainty ($\epsilon$) over time.

et al. [40] discussed "algorithmic extrapolation" where their networks solved problems of greater difficulty than those in their training set by performing additional computations. Graves [37] proposed the use of a "halting unit" whose activation level determined the probability of continuing model computations. Finally, Figurnov et al. [43], motivated by [37], adaptively determined the spatial extent and number of processing layers in a deep convolutional network. In a broader sense, "conditional computing" approaches aim to learn dropout-like policies [44, 45] to train neural networks that can balance parsimony and performance. While the approaches presented in [37, 43–45] are impressive advances, they neither operate explicitly with cRNNs nor yield RT values for downstream comparisons on visual cognitive tasks. The elegant approach of [26] suffers from the primary drawback that it relies on the backpropagation through time (BPTT) algorithm. BPTT imposes high memory demands, limiting the number of timesteps a cRNN can perform and therefore condemning any derived measure to be coarse. Furthermore, vanilla BPTT forces the dynamics to be constant (i.e., the cRNN arrives at a solution at $t = T$ for any input) making it non-stimulus-adaptive [8].

**Probabilistic interpretation of model uncertainty.** Decision certainty and computational parsimony are duals of each other. In other words, easier tasks require fewer computational resources, with models typically being most certain about their predictions, while the opposite holds for harder tasks. Modeling decision uncertainties provide another effective angle to capture the "computational demand" of a stimulus condition. While deterministic neural networks suffer from ill-calibrated prediction confidence in Softmax outputs, Bayesian neural networks derive prediction confidence from their weight uncertainties but are generally not performant. Inspired by the Dempster-Shafer theory of evidence [29], Sensoy et al. [30] propose a formulation that treats model outputs as parameters of a Dirichlet distribution that signifies the total amount of evidence accumulated for each class. Although influential, this body of work, called Evidential Deep Learning (EDL), has never been linked to RT measures or applied to visual cognitive tasks. We refer the interested reader to Ulmer et al. [46] for a comprehensive survey of this literature. In this paper, we specifically leverage EDL in conjunction with attractor dynamics to support the derivation of an RT proxy.

## 3 General Methods

### 3.1 Models

**cRNN.** The model of interest chosen in this work is the horizontal gated recurrent unit (hGRU; [19]), a convolutional recurrent neural network model that we will canonically refer to as cRNN throughout the rest of the paper. In principle, our framework applies to other cRNN architectures as well. The hGRU is inspired by the anatomical and neurophysiological properties of the primate

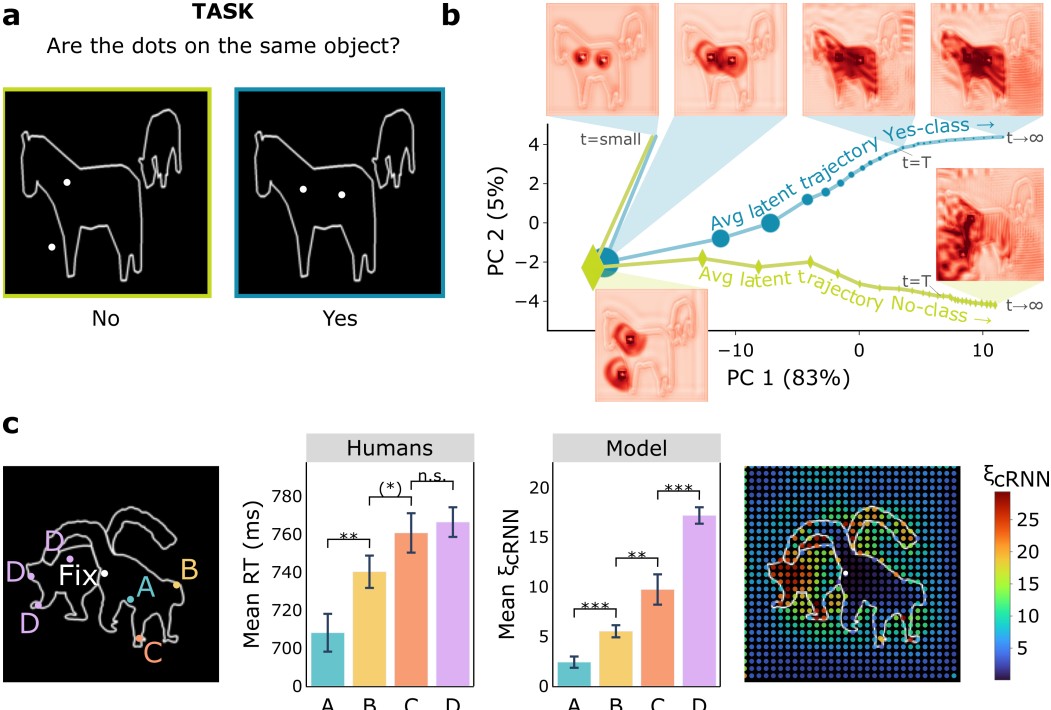

Figure 2: **Human versus cRNN temporal alignment on an incremental grouping task. a.** Description of the task (inspired by cognitive neuroscience studies [47]). **b.** Visualization of the cRNN dynamics. The two lines represent the average latent trajectories across $1K$ validation stimuli labeled "yes" and "no" respectively. Marker size indicates average uncertainty $\epsilon$ across the same stimuli. Evident from the graph is that the two trajectories start to diverge after some initial time passes, clearly separating the two classes. Owing to the C-RBP training algorithm and attesting to the dynamics approaching an equilibrium, the step sizes become increasingly small over time. We also include snapshots of the latent activity in the cRNN for two example stimuli (one "yes", one "no"; see Panel a) along the respective trajectory. Each snapshot represents $\mathbf{h_t}$ at time step $t$, averaged across the channel dimension and smoothed by averaging $\mathbf{h_t}$ and $\mathbf{h_{t+1}}$. Notice the spread of activity over time. The cRNN gradually fills the segment containing the dots. The strategy can be appreciated even better in video format (see SI §D). **c.** Comparison against data from the experiment with human participants in [47]. The position of one dot (white, labeled "Fix") was kept constant, while the position of the other was manipulated to create experimental conditions of varying difficulty. These manipulations have qualitatively similar effects on the cRNN as they do on human behavior. Error bars represent the standard error of the mean across stimuli. The significance levels for shown contrasts were adjusted downward from .1 (*), .05*, .01**, and .001*** using Bonferroni. The spatial uncertainty map shown on the right visualizes the spatial anisotropy observed in the model for the same example stimulus shown on the left.

visual cortex. Architectural facets of the hGRU include an hypercolumnar organization, distinct excitatory and inhibitory subpopulations, and local- and long-range feedback mechanisms, rendering it overcomplete and an ideal candidate cRNN for our purposes. The hGRU and its extensions have previously demonstrated proficiency in learning long-range spatial dependency challenges, including naturalistic ones [8, 19, 48]. For implementational details, refer to SI §A.

**Control models.** To further motivate our cRNN choice, we refer to its ability to systematically generalize to more challenging task settings after being trained on less challenging ones [8]. Inference-time flexibility and performance are critical desiderata, given that our goal is to probe the computational demands of a task via stimulus manipulations. We first demonstrate that the cRNN's performance remained well above chance on exceedingly challenging versions of the first task we consider (see Section 4.1) after training on a simplified version (see SI §B for full experimental details). We trained pure feedforward convolutional neural networks (CNNs) as control models under identical training

data distributions. The CNNs struggled to generalize effectively, as did a naive control RNN model (Fig. S1). Because CNNs lack explicit temporal dynamics and our recurrent control models failed our generalization test, we limit the experimental results presented in this paper to our cRNN. For completeness, we remark that a Vision Transformer [49] passed the generalization test. However, we exclude these models from our analyses for two reasons. First, as such, Vision Transformers lack biological analogs. Second, our focus in this work is to derive RT measures from model dynamics, which transformers do not exhibit.

## 3.2 Training Framework

**General remarks.** All model runs reported herein were done from *scratch* using only the ground truth class labels for supervision. Any alignment between the cRNN model and humans would have had to emerge as a byproduct of task constraints. We emphasize that no human RT data were involved in our training routines. For the cRNNs, we computed a task loss only at the network's end state to preserve the ability to remain flexible during inference. The cRNNs were trained in a data-parallel manner on 4 Nvidia RTX GPUs (Titan/3090/A5000) with 24GB of memory each. Each training run took approximately 48 hours, except those in Section 7 (2 hours), which were faster due to the smaller training dataset size. They were furthermore trained with Adam [50], a learning rate of $1e-3$, and $\gamma = 100$ for the C-RBP penalty ([8]; also see SI §A). We use $T = 40$ recurrent timesteps during the training phase, with an exception for the task in Section 6, where $T = 80$ and $\gamma = 1000$.

**Stabilizing cRNNs.** We train our cRNN model with the contractor recurrent back-propagation algorithm (C-RBP; [8]). C-RBP imposes an attractor-dynamic constraint on the cRNN's computational trajectory, imparting several benefits over the standard BPTT algorithm. C-RBP is significantly more memory efficient and affords arbitrary precision in the cRNN temporal dynamics (i.e., $T$ can be high). C-RBP also allows the cRNN to learn stable and expressive solutions. We can observe this from the cRNN's latent trajectories, as visualized in Fig. 2b, where extrapolation in time (beyond $T$) is non-catastrophic. We argue that model instability is why BPTT-trained controls failed the generalization test (Fig. S1). Furthermore, stable solutions imparted by C-RBP allow our cRNNs to dynamically adapt their number of processing steps in a stimulus-dependent manner. That is, we pose that a C-RBP-trained cRNN can utilize its computational budget differently depending on the input (see Fig. S2 for an illustration). Finally, we note that our strategy to achieve stable recurrent solutions is in contrast to that employed by Bansal et al. [40], wherein cRNNs were stabilized through architectural choices along with an incremental training process.

**Evidential loss.** We augment our cRNN training objective with an evidential deep learning loss (EDL; [30]). In contrast to the classic cross-entropy loss, EDL considers cRNN outputs to be the parameters of a belief distribution rather than point estimates of class probabilities (Fig. 1a). We formalize cRNN beliefs as a Dirichlet distribution. Strong evidence in favor of one class causes the Dirichlet distribution to be peaked, reflecting low uncertainty ($\epsilon$). In contrast, when there is a lack of overall evidence, the Dirichlet distribution becomes more uniform, thereby increasing $\epsilon$. For a complete mathematical specification of this loss formulation, refer to SI §A.

## 3.3 RT Metric ($\xi_{cRNN}$)

We derive a novel metric to quantify the differential computational demands imposed on the cRNN by stimulus manipulations and capture the essence of model decision time courses for alignment. Modeling uncertainty ($\epsilon$) explicitly allows us to track its evolution over time from the cRNN (Fig. 1b). We formulate our RT metric ($\xi_{cRNN}$) as the area under this uncertainty curve. Precisely, $\xi_{cRNN} = \int_0^T \epsilon(t)\, dt$ (Fig. 1b). Intuitively, high (low) $\xi_{cRNN}$ can be understood as the cRNN spending a large (small) amount of time in an uncertain regime. Our metric affords direct comparisons to human RTs. In the following sections, we apply this metric to cRNNs trained on a host of visual cognitive tasks and study their alignment to human reaction time data.

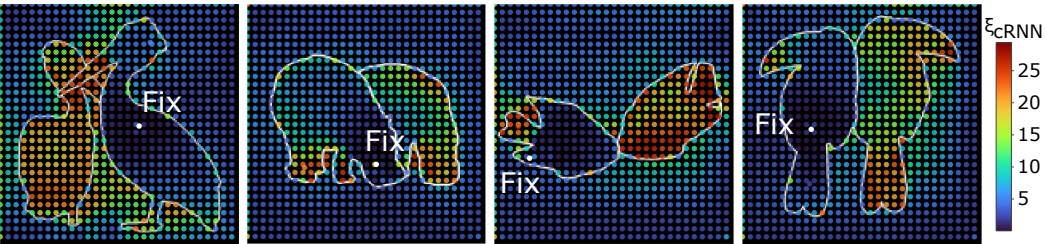

Figure 3: $\xi_{cRNN}$ **is spatially anisotropic.** We present spatial uncertainty (SU) maps to visualize the impact a given fixed dot position will have on model RTs when the position of the second dot is varied. Higher (lower) values of $\xi_{cRNN}$ represent slower (faster) model responses.

## 4 Incremental Grouping

### 4.1 Task, Stimuli, and Visualizations

Inspired by an incremental perceptual grouping task in humans [47], we propose an equivalent challenge for the cRNN where the model is required to determine whether two dots are on the same object (Fig. 2a). The stimuli consist of white outlines of natural shapes on a black background ($150 \times 150$ px). We created $340K$ ($15K$) training (validation) outline stimuli from MS COCO images [51]. The dots were placed semi-randomly such that there was no difference in the distribution of dot distances between the two stimulus classes.

We additionally introduce two novel types of visualizations to probe cRNN visual strategies. First, we track cRNN dynamics by looking at its latent activity $\mathbf{h_t}$ at every time step $t$, averaged across the channel dimension and smoothed by averaging $\mathbf{h_t}$ and $\mathbf{h_{t+1}}$. To facilitate interpretation, we provide these videos for the reader (see SI §D). Second, we compute a "spatial uncertainty" (SU) map for each outline stimulus by keeping one dot fixed and varying the position of the other. Each spatial coordinate has a value equal to $\xi_{cRNN}$ for that corresponding dot configuration. Examples are shown in Fig. 2c (right) and Fig. 3.

### 4.2 Results

Despite its lightweight architecture, our cRNN solved the incremental grouping task, achieving $98\%$ validation accuracy. Visualizing the cRNN latent activities revealed that the model learned a cognitively-viable filling-in strategy. Network activity originated at each of the dot locations and gradually spread, eventually filling up the whole object(s) (Fig. 2b). We note that this behavior is an emergent property from purely the classification task constraint instead of direct supervision for any form of segmentation. We then sought to ask if $\xi_{cRNN}$ effectively captures these visual decision-making dynamics. In the following sections, we report results detailing the relationship between $\xi_{cRNN}$ and a host of complex, topological stimulus properties.

$\xi_{cRNN}$ **captures the serial spread of attention in our models.**    Human visual strategies in solving the incremental grouping task are thought to involve a serial spread of attentional resources, resulting in longer RTs for higher distances [47]. To test this effect in our model, we took a subset of our validation outlines and repositioned the dots to be in the center of an object, 14 px apart. We then systematically kept moving the dots 14 px more apart until reaching the object boundary. A linear mixed-effects regression analysis revealed a significant positive effect of this manipulation on $\xi_{cRNN}$, $b = 1.46, SE = 0.11, z = 13.58, p < .001$ (Fig. S3, Fig. S4). The SU maps (Fig. 2c right, Fig. 3) exhibit this distance effect too. Placing the second dot further away from the fixed dot (indicated in white) yielded higher $\xi_{cRNN}$ values.

$\xi_{cRNN}$ **recapitulates the spatial anisotropy in human RT data.**    Euclidean distance only paints a partial picture. Human RTs are affected by several other factors, such as topological distance, the proximity of the dots to boundaries, and the need to spread through narrow object regions, all amounting to a well-documented spatial anisotropy in the spread of visual attention [47, 52]. While anisotropy is evident in the SU maps from our cRNN (Fig. 2c right, Fig. 3), we also formally assess the alignment between humans and the cRNN in this regard, using human RT data from [47].

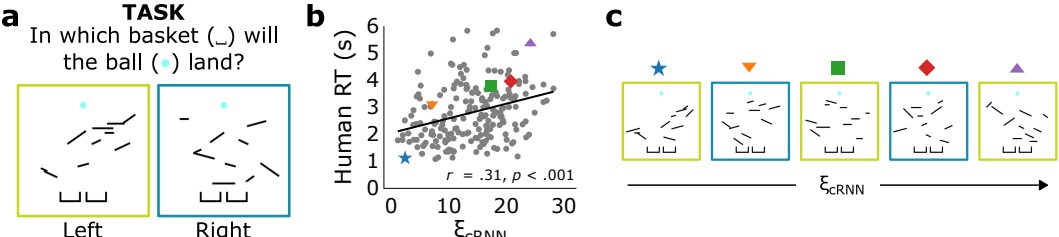

Figure 4: $\xi_{cRNN}$ **from our model recapitulates human RTs on the Planko task. a.** The stimuli consist of a ball on top, followed by ten randomly placed planks and finally two baskets in the bottom. Participants are shown these static stimuli and are asked to guess if the ball will end up in the left or the right basket when released and allowed to bounce around on the planks. **b.** Human RTs (collected in [53]) are predicted by $\xi_{cRNN}$. The five colored symbols reflect the data points obtained from the stimuli shown in the next panel. **c.** Five example stimuli of increasing $\xi_{cRNN}$.

First, Jeurissen et al. [47] quantified their stimuli on an alternative dot distance measure that went beyond Euclidean distance to reflect the other aforementioned factors and was more predictive of human RT (see SI §F). We found this measure to be a better predictor of $\xi_{cRNN}$ ($r = .80$) as well (versus $r = .45$ for Euclidean distance), $t(63) = 4.68, p < .001$. Second, the stimulus set featured distinct experimental conditions, varying in expected difficulty based on the position of the dots (Fig. 2c left). As shown in Fig. 2c (middle), the effects of this manipulation on $\xi_{cRNN}$ were qualitatively aligned with those on human RT. A one-way repeated measures ANOVA across 22 outline stimuli revealed a significant effect of "Condition" on $\xi_{cRNN}$, $F(3, 63) = 67.65, p < .001$, as well as on the average RTs from [47], $F(3, 63) = 16.95, p < .001$. Furthermore, Fig. 2c (middle) presents the results of planned contrasts (paired t-tests). $\xi_{cRNN}$ was significantly higher for B compared to A, $Cohen's\ d = 2.00, t(21) = 9.39, p < .001$, but also for the more difficult Condition C compared to B, $d = 0.73, t(21) = 3.42, p = .003$. Third, we observed a significant overall correlation between human RT and $\xi_{cRNN}$ on Yes-stimuli, $r = .31, p = .01$. Fourth, on a novel test set of our own, paired t-tests showed that $\xi_{cRNN}$ was significantly higher when two dots on the same object were placed in narrow versus wide object regions, $d = 0.70, t(22) = 3.34, p = .003$, and when the path was curved (i.e., longer topological distance) versus when it was not, $d = 0.98, t(22) = 4.71, p < .001$ (see SI §G, Fig. S5). Finally, SI §G also describes an experiment in which we varied the distance between one dot and the object boundary while keeping the distance between the dots themselves constant. We found that our model exhibits higher values near the boundaries, which is strikingly similar to human behavioral results presented in [47].

**Ambiguous stimuli.** As a final way of gaining insight into the cRNN's task behavior, we looked at those stimuli in the validation set that elicited the highest $\xi_{cRNN}$ values. We observed that some were characterized by uncertainty curves that remain high, even at $t = T$, suggesting the model deems them ambiguous. In a test set we designed to be ambiguous (it features occlusions), we frequently observed such non-decreasing uncertainty curves too (see SI §H). We note that while our training procedure promotes stable attractor states in the model's *latent* dynamics, this is disparate from readout dynamics allowing the framework to truly explore the spectrum of computation times and uncertainties.

Taken together, these results suggest that our metric is able to capture the dynamical signatures of object-based attention in our cRNNs and provides a means to quantify alignment with human visual strategies.

## 5 Visual Simulation - *Planko*

### 5.1 Task and Stimuli

The Planko task (and corresponding dataset) was introduced in [53] and features stimuli ($64 \times 64$ px; $80K$ training, $5K$ validation) displaying a ball about to fall down and eventually land in one of two baskets ("left"/"right"). The outcome depends on the position and orientation of a set of planks placed in between the ball and the baskets (Fig. 4a). Humans are thought to solve this task through mental

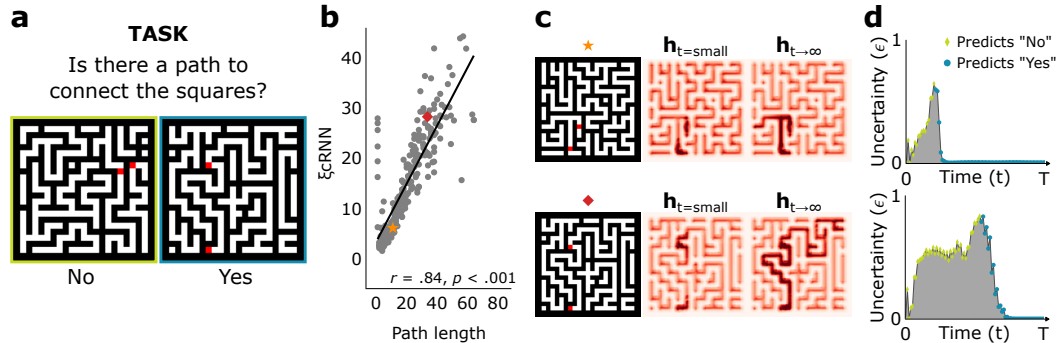

Figure 5: **cRNN learns filling-in strategy to solve mazes. a.** Task description. **b.** The cRNN model takes more time to solve mazes with longer path lengths, an effect previously found in human participants too. **c.** Latent activity visualizations for two yes-mazes. The cRNN gradually fills the segment containing the cues. The strategy can be appreciated even better in the videos supplied in the SI. **d.** Uncertainty curves for the two inputs shown in Panel c. The cRNN remains uncertain for much longer for the maze featuring the longest path. The uncertainty evolution is visualized dynamically in conjunction with the change in latent activity in the supplementary videos.

visual simulation [53]. Notably, feedforward CNNs struggle to solve the task [54] and when they do, they appear to adopt non-aligned strategies to humans [53]. In contrast, cRNNs show evidence of simulation-like behavior on a variant of the task [54]. With our metric, we are now able to directly compare human (collected by [53]) and cRNN RTs. To this end, we trained a cRNN from scratch on Planko (see Section 3.2 and SI §A for details on training).

### 5.2 Results

The stimuli displayed in Fig. 4 illustrate the challenge of the Planko task. Our cRNN achieves a performance of $84\%$ on the validation set, well within the confidence interval of human accuracy [53]. Stimuli presented to human participants in the original study ($N = 200$) were used here as a held-out set. We observed a significant correlation between model $\xi_{cRNN}$ values and human RTs, $r = .31, p < .001$) (Fig. 4b). This result indicates that stimuli that were harder for humans tended to be computationally challenging for the cRNN (see examples in Fig. 4c).

*What makes a stimulus challenging?* Variance in human RTs is known to be partially explained by the statistics of the planks. For instance, stimuli for which the outcome is more likely to change as a consequence of perturbing the plank configurations by a small amount were deemed "uncertain". Indeed, participants responded slower to these stimuli [53]. We were able to replicate this result in our cRNN, $r = .31, p < .001$. These results showcase how our methodology can be used to make stimulus-dependent RT predictions. This could help shed light on, for example, the remarkable yet under-examined ability of humans to flexibly adopt different strategies in a Planko task depending on the stimulus demand.

## 6 Mazes

### 6.1 Task and Stimuli

We devised a maze task where the goal is to tell whether two cued locations are connected by a path (Fig. 5a). To build our training ($N = 34.5K$) and validation ($N = 14K$) set, we started from mazes provided by [40] but modified them to make them suitable for classification (the originals did not feature any disconnected paths). The cues were placed such that the two classes ("yes"/"no") did not differ in Euclidean cue distance. Each maze was $144 \times 144$ px.

### 6.2 Results

Here too, we observed that the uncertainty curves and derived $\xi_{cRNN}$ values differed considerably depending on the input, even if nearly all mazes were classified correctly at $t = T$ ($99\%$ correct

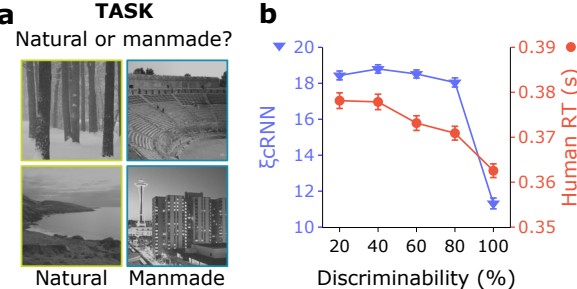

Figure 6: $\xi_{cRNN}$ **is predictive of the discriminability of a visual scene. a.** Task description. **b.** Human RTs (collected in [57]) decrease as a function of discriminability, as do $\xi_{cRNN}$. Highly discriminable scenes are easier for both human participants and the cRNN model. Error bars represent the standard error of the mean.

on validation mazes). For the Yes-mazes contained in a random subset of the validation set, we found that the model spent more time being uncertain (higher $\xi_{cRNN}$) when the length of the path between the two cued locations was longer, $r = .84, p < .001$ (Fig. 5b,d). Moreover, path length was a significantly better predictor than simple Euclidean distance ($r = .39$), $t(234) = 5.81, p < .001$. Similarly, human RTs on maze tasks have been shown to increase with path length [55]. When the cued locations were on two distinct maze segments (No-mazes), $\xi_{cRNN}$ was significantly predicted by the mean of both segment lengths, $r = .56, p < .001$. The kind of visual strategy that produced such distance effects can be understood by inspecting the visualizations of the latent activities (Fig. 5c). The model gradually spread activity along the maze segment(s) containing the cues. We strongly encourage the reader to watch our videos (see SI §I) to better appreciate this spread in conjunction with the evolution of uncertainty over time. Finally, we did not find evidence for an effect on $\xi_{cRNN}$ of the number of T-junctions on the path, which are known to predict human RT [56]. We speculate that this potential misalignment, worth investigating in future work, could be due to the model not being restricted in how many possible paths it can explore at once.

## 7 Scene Categorization

### 7.1 Task and Stimuli

To showcase how our general approach can also be applied to study higher-level visual tasks, we consider a natural versus manmade scene categorization task as our final challenge (Fig. 6). Human RT data from a rapid-scene categorization version of this task were already available from [57]. We selected images from the SUN database [58] querying the same classes as in [57], but excluding the images shown to human participants, and performed a stratified train ($N = 8.8K$) and validation ($N = 979$) split. The image preprocessing pipeline entailed cropping the image to be square, resizing to $150 \times 150$ px, converting to grayscale, and random horizontal flipping. In addition, the images were matched for overall brightness.

### 7.2 Results

The model was able to correctly classify $83\%$ of the validation images and $73\%$ of the held-out test images from [57]. The errors can be understood, at least partly, in light of the less-than-clear-cut labels (e.g., an islet with a house on top, a building behind many trees). Each of the test images had previously been assigned a discriminability score [57], based on the average error rate of logistic classifiers trained on "gist" features [59]. Human participants were reported to be slower with decreasing discriminability [57]. Here, we show a similar trend in the model (Fig. 6b), with a linear regression analysis revealing a significant, negative discriminability slope, $b = -0.47, SE = 0.03, t = -16.45, p < .001$. Directly comparing $\xi_{cRNN}$ to human RT, we found them to be significantly correlated, $r = .19, p < .001$. While our cRNN model of choice here does not perfectly account for empirical patterns in human reaction times, specifically in their non-monotonic reaction time readouts for the low-discriminability stimuli, they provide a promising

start. Our framework is model-agnostic (e.g., see SI §J for results on a convLSTM) and thus can be used to test hypotheses on architectural improvements.

## 8 Discussion

We are in the midst of a paradigm shift in visual neuroscience. The continued development of computational models that rival human behavioral performance has called for tools to quantify how well the internal mechanisms of such models compare to biological truths. These efforts have demonstrably been beneficial for both the AI and neuroscience ecosystems.

In this work, we push this frontier by introducing a framework to characterize the evidence accumulation strategies of stable recurrent vision models. Our experiments thus far show the potential to generate novel hypotheses of interest to visual neuroscientists. The most direct application of our framework for experimental design is in taxonomizing stimuli in terms of the putative computational demand based on our metric toward understanding neural processes at varying timescales. Furthermore, the value proposition of this framework includes the ability to adjudicate between models based on the consistency of their latent dynamics with human reaction time data. For example, our metric from a cRNN trained on the maze task is uncorrelated with the number of T-junctions in the stimulus, indicating a potential divergence between serial and parallel exploration strategies (SI §I). It is worth noting that distinguishing between serial and parallel attention models has been of long-standing interest [60]. Similarly, testing a zoo of hierarchical models on the scene categorization task while keeping an eye on the degree of alignment with our metric can potentially lend novel mechanistic insights and help delineate bottom-up, lateral, and top-down processes [61]. We are also enthusiastic about drawing theoretical parallels between our metric summarizing the cRNN's evidence accumulation strategy and the complexity of natural data statistics toward understanding the sample efficiency of learning in models and primates.

**Limitations and an outlook for the future.** Within the scope of this work, we have considered four disparate visual cognition tasks to showcase how our approach offers the means to study the temporal alignment between recurrent vision models and the human visual system. Of course, in the face of the myriad of tasks that cognitive scientists have used over centuries, our selection still needs to be improved. Our results do not directly speak to the diversity in response types, stimulus types, or naturalism, to name a few. However, our approach is general-purpose enough to adapt to newer tasks as long as they are cast as a classification problem. We believe this should cover a significant share of cognitive science and AI tasks. Extending the approach beyond choice tasks would require a considerable revision of how to make the model's uncertainty explicit. Another limitation is that there could easily be other cRNNs that align with human RTs better. While we motivate the choice for the cRNN in Section 3.1, our main contribution here is to introduce the overall framework integrating stable recurrent vision models with evidential learning theory to support the derivation of a novel RT metric. A comprehensive comparison of different models, apart from those in SI §B, was hence beyond this work's scope. However, we are confident our framework paves the way to build a taxonomy of models in the future as well as generate testable hypotheses for neuroscience.

**Broader impacts.** With AI vision models becoming ubiquitous in our daily lives, the call to make them fairer, transparent, and robust is growing louder. Our framework facilitates this by allowing researchers to peek into the inner working of these AI models for transparency and thinking about their match to biological systems, which are arguably more robust. From a neuroscience perspective, computational models of RT are gaining prominence in computational psychiatry and hold the key to understanding many neurobiological disorders [62, 63]. We generally do not anticipate any negative impact on society in either scenario.

## Acknowledgments

The authors would like to explicitly thank Jeurissen et al. [47] and Sofer et al. [57] for generously sharing their stimuli and human reaction time data, and Bansal et al. [40] for their maze stimuli. We are also grateful to Drew Linsley for his helpful comments and feedback. This work was supported by ONR (N00014-19-1-2029), and NSF (IIS-1912280). Additional support provided by the Carney

Institute for Brain Science and the Center for Computation and Visualization (CCV). We acknowledge the computing hardware resources supported by NIH Office of the Director grant S10OD025181.

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

# Supplementary Information

## A Implementation Details

### A.1 Model Implementation

We instantiate our cRNN as a canonical recurrent convolutional neural network layer inspired by the primate visual cortex's anatomical and neurophysiological properties. This model, termed hGRU and first introduced in Linsley et al. [19], implements an hypercolumnar organization with distinct excitatory and inhibitory subpopulations and local- and long-range feedback mechanisms. For the sake of completeness, we outline the governing equations of the model below.

First compute suppressive interactions:

$$\mathbf{G}^S = sigmoid(U^S * \mathbf{H}[t-1]) \qquad \text{\# Compute suppression gate as function of current state}$$
$$\mathbf{C}^S = BN(W^S * (\mathbf{H}[t-1] \odot \mathbf{G}^S)) \qquad \text{\# Applying the gate}$$
$$\mathbf{S} = \left[\mathbf{Z} - \left[(\alpha\mathbf{H}[t-1] + \mu)\,\mathbf{C}^S\right]_+\right]_+, \qquad \text{\# Additive and multiplicative suppression of } \mathbf{Z}$$

Then compute the facilitation terms followed by the recurrent update:

$$\mathbf{G}^F = sigmoid(U^F * \mathbf{S}) \qquad \text{\# Facilitation gate}$$
$$\mathbf{C}^F = BN(W^F * \mathbf{S}) \qquad \text{\# Facilitation term}$$
$$\tilde{\mathbf{H}} = \left[\nu(\mathbf{C}^F + \mathbf{S}) + \omega(\mathbf{C}^F * \mathbf{S})\right]_+ \qquad \text{\# Modulation of } \mathbf{S} \text{ through additive and multiplicative facilitation}$$
$$\mathbf{H}[t] = (1 - \mathbf{G}^F) \odot \mathbf{H}[t-1] + \mathbf{G}^F \odot \tilde{\mathbf{H}} \qquad \text{\# State update}$$

In these equations, $\mathbf{H}, \mathbf{Z} \in \mathcal{R}^{X \times Y \times C}$ represent the cRNN's latent state and bottom-up feedforward drive, respectively, with height/width/channels $X, Y, C$. $W^S, W^F \in \mathbb{R}^{E \times E \times C \times C}$ are learnable kernels controlling the suppressive and facilitatory interactions and $E$ is the spatial width of the lateral interactions. For all our experiments, we set $E = 5$, except for Planko where $E = 3$. Similarly, $U^S, U^F \in \mathbb{R}^{1 \times 1 \times C \times C}$ are learnable kernels controlling the gates. Model dynamics are discretized, with $t \in [1...T]$ representing the timesteps of processing. We use *softplus* as our rectifying nonlinearity (denoted by $[\cdot]_+$). We also use Batch Normalization (*BN*) in the model to control exploding/vanishing gradients [64]. *BN* kernels are shared across timesteps of processing.

### A.2 Training Implementation Details

**Leveraging equilibrium dynamics for training.** Training cRNNs has traditionally posed a problem due to the severe memory bottlenecks imposed by the backpropagation through time (BPTT) algorithm. To remedy this, Linsley et al. [8] introduced a variant of the classic recurrent backpropagation algorithm (RBP) that leveraged equilibrium dynamics to compute model gradients with $O(1)$ memory complexity. We outline the high-level idea here and refer the reader to [8] for more details.

Our discretized cRNN dynamics can be written as follows,

$$\mathbf{H}[t] = f(\mathbf{H}[t-1], \mathbf{X}; \mathbf{W}),$$

where $f(.)$ is the model architecture described, $\mathbf{H}[t]$ is the cRNN's state at timestep $t$, and $\mathbf{X}$ is the bottom-up sensory input. For convenience, we use $\mathbf{W}$ to denote all the trainable parameters of $f(.)$. If $f(.)$ is a contraction mapping, then $\mathbf{H}[t] \to \mathbf{H}^*$ as $t \to \infty$. One can define an implicit function as follows to aid in the gradient computation: $\psi(\mathbf{H}; \mathbf{X}, \mathbf{W}) = \mathbf{H} - f(\mathbf{H}, \mathbf{X}; \mathbf{W})$. By construction, we can observe that $\psi(\mathbf{H}^*; \mathbf{X}, \mathbf{W}) = 0$ and by extension,

$$\frac{\partial\psi(\mathbf{H}^*; \mathbf{X}, \mathbf{W})}{\partial\mathbf{W}} = 0$$

Following a sequence of algebraic manipulations we can compute the gradient of the equilibrium state $\mathbf{H}^*$ w.r.t. parameters $\mathbf{W}$ as follows.

$$\frac{\partial \mathbf{H}^*}{\partial \mathbf{W}} = (I - J_{f,\mathbf{H}^*})^{-1} \frac{\partial f(\mathbf{H}^*, \mathbf{X}; \mathbf{W})}{\partial \mathbf{W}},$$

where $J_{f,\mathbf{H}^*}$ is the Jacobian of the cRNN $f(.)$ evaluated at $\mathbf{H}^*$. The ability to compute this gradient, however, relies on $(I - J_{f,\mathbf{H}^*})$ being invertible. To guarantee this, Linsley et al. [8] introduced the Lipschitz Coefficient Penalty (LCP) as $\mathcal{L}_{LCP}(\mathbf{W}) = \|(\mathbf{1} \cdot J_{f,\mathbf{H}^*} - \lambda)^+\|_2$ which controls the degree of contraction in $f(.)$ and by extension the invertibility of the term discussed above. Moreover, this regularization can be added to any task loss which we exploit.

**Evidential learning formulation.**     To support our goal of characterizing evidence accumulation strategies in our cRNN, we parameterize our model readout to explicitly express uncertainty. Specifically, we follow the evidential learning strategy introduced in Sensoy et al. [30] to interpret cRNN readouts as the parameters of the Dirichlet density on the class predictors, $\boldsymbol{\alpha} = < \alpha_1, ..., \alpha_K >$, where $K$ is the total number of classes in the classification problem of interest. As in [30], our evidential loss comprises of two components, one that supports belief accumulation for the "correct" class ($\mathcal{L}_{risk}$) and a second that ensures that the beliefs for all the other "wrong" classes have uniform mass ($\mathcal{L}_{balance}$).

$$\mathcal{L}_{EDL}(\mathbf{W}) = \sum_{i=1}^{i=N} \mathcal{L}_{risk}^{(i)} + \rho \mathcal{L}_{balance}^{(i)}(\mathbf{W})$$

$$\mathcal{L}_{risk}(\mathbf{W}) = \sum_{j=1}^{j=K} (y_j - \hat{p}_j)^2 + \frac{\hat{p}_j(1 - \hat{p}_j)}{S + 1}$$

$$\mathcal{L}_{balance}(\mathbf{W}) = D_{KL}(D(\mathbf{p}|\hat{\boldsymbol{\alpha}}) \, \| \, D(\mathbf{p}| < 1, ..., 1 >))$$

In the above, $\mathbf{y}$ denotes a one-hot ground truth label for a given sample with $y_j = 1$ if the sample belongs to class $j$, and 0 otherwise. $\hat{p}_j$ is from the model that denotes the expected probability of the $j$-th singleton under the corresponding Dirichlet such that $\hat{p}_j = \frac{\alpha_j}{S}$. $S = \sum_{j=1}^{K} \alpha_j$ resembles the "total evidence" accumulated by the model (or the Dirichlet strength). $D_{KL}$ is the Kullback–Leibler divergence measure. $\hat{\boldsymbol{\alpha}}$ are the Dirichlet parameters of just the "misleading" evidence given by $\hat{\boldsymbol{\alpha}} = \mathbf{y} + (1 - \mathbf{y})\boldsymbol{\alpha}$. We define the instantaneous uncertainty as $\epsilon = \frac{K}{S}$.

Finally, $\rho$ is an annealing term that is used to balance the relative importance of $\mathcal{L}_{risk}$ and $\mathcal{L}_{balance}$. In practice, we use the following increment schedule: $\rho_{epoch} = min\left[1, \frac{epoch}{\tau}\right]$, where $\tau$ is a hyperparameter. For a complete derivation of the loss formula, we refer the reader to [30]. We choose $\tau = 16$ for incremental grouping, $\tau = 10$ for Planko, $\tau = 20$ for the maze task, and $\tau = 16$ for scene categorization.

**Training objective.**     Our final training objective is thus $\mathcal{L}(\mathbf{W}) = \mathcal{L}_{EDL}(\mathbf{W}) + \gamma \mathcal{L}_{LCP}(\mathbf{W})$.

# B   Generalization Experiments

## B.1   Task and Stimuli

In the main text, we identified the flexibility to generalize to more complex task settings at inference as a critical desideratum for obtaining a human-like RT metric from any model. We tested generalizability in the cRNN as well as in a set of control models (see further) using the incremental grouping task described in the main text (dots on the same object "yes"/"no"?). We trained on an easy version of the task by only selecting training stimuli for which the Euclidean dot distance was at most $44$ px ($50^{\text{th}}$ percentile). We refer to this reduced training set as "short training stimuli" ($N = 170K$). For evaluation, we created two subsets of the validation set, one easy and one hard. The easy set ("short validation stimuli", $N = 7.7K$) consisted of those validation stimuli for which the dots were at most $44$ px apart. The hard set ("long validation stimuli", $N = 1.5K$) consisted of those stimuli for which the dots were more than $82$ px apart ($90^{\text{th}}$ percentile). All thresholds were computed across the training stimuli in the original set.

## B.2   Models and Training

All models listed below were trained from scratch with a batch size of $128$ and a standard cross entropy loss. Training was done in a data-parallel manner on four Nvidia RTX GPUs (Titan/3090/A5000) with 24GB of memory each.

- **AlexNet-BN**. This model was included as a pure feedforward control. We used the PyTorch implementation for AlexNet [65] with *BN* provided in [66]. The optimizer was stochastic gradient descent (SGD) [67] and the learning rate was $1e-2$. We report the results for the best epoch out of $59$ (based on performance on the short validation stimuli) for three individual runs.

- **VGG-19-BN**. This model was included as an additional pure feedforward control. We trained a variant of the original VGG architecture [68] referred to as "vgg19_bn" in the torchvision model zoo [69]. We used an SGD [67] optimizer and a learning rate of $1e-2$. We report the results for the best epoch out of $32$ (based on performance on the short validation stimuli) for three individual runs. Interestingly, in two additional runs with the same hyperparameters, VGG-19-BN was unsuccessful in solving even the easy version of the task and performed at chance level.

- **hGRU-BPTT**. This is a recurrent control model with the same architecture (see SI §A) as the model studied in the main paper (referred to there as cRNN), but trained with the classic BPTT. Furthermore, we found that in order to learn the easy version of the task with BPTT, we had to replace *BN* with group normalization. The optimizer was Adam [50], $lr = 1e-3$, and $T = 15$. Our hypothesis was that this control model would fail the generalization test and perform near chance level on the harder long validation stimuli. We report the results for the best epoch out of $32$ (based on performance on the short validation stimuli) for three individual runs.

- **hGRU-CRBP**. This model shares the same recurrent architecture (see SI §A) and training algorithm algorithm (C-RBP) as the cRNN used in the main paper. We trained the hGRU-CRBP models with an Adam [50] optimizer, $lr = 1e-3$, $T = 40$, and $\gamma = 100$. Our hypothesis was that this model would still perform well above chance level on the harder long validation stimuli, in contrast to both the feedforward controls and the RNN control without C-RBP. This hypothesis was motivated by findings in [8]. We report the results for the best epoch out of $32$ (based on performance on the short validation stimuli) for three individual runs.

- **ConvLSTM**. This is a recurrent control model with the LSTM architecture [70] that shares the C-RBP training algorithm used to train the cRNNs in the main text, but lacks the anatomical and neurophysiological constraints. We trained the ConvLSTM models [71] with Adam [50], $lr = 1e-3$, $T = 40$, and a $\gamma$ of either $100$ or $1000$. We analyzed the generalization performance of the runs that reached at least $90\%$ accuracy on the short validation stimuli.

- **ViT-B-16**. ViT-B-16 was included as a control model from the Vision Transformer family [49]. We used the implementation included in the torchvision zoo [69] as "vit_b_16". To

train the model, we used an Adam [50] optimizer, a patch size of 10, a learning rate of $1e-2$, a dropout rate of .10, and an attention dropout rate of .10. We report the results for the best epoch out of 299 (based on performance on the short validation stimuli).

## B.3 Results

Fig. S1 summarizes the results of the generalization experiments. The models that share the same architecture and training algorithm (hGRU-CRBP) as the cRNNs studied in the main text performed well above chance on the long validation stimuli. Despite having high accuracy on the short validation stimuli (i.e., those with dot distances within the range they were trained for), the control models performed near or at chance on the harder long validation stimuli. The only exception to this finding was ViT-B-16.

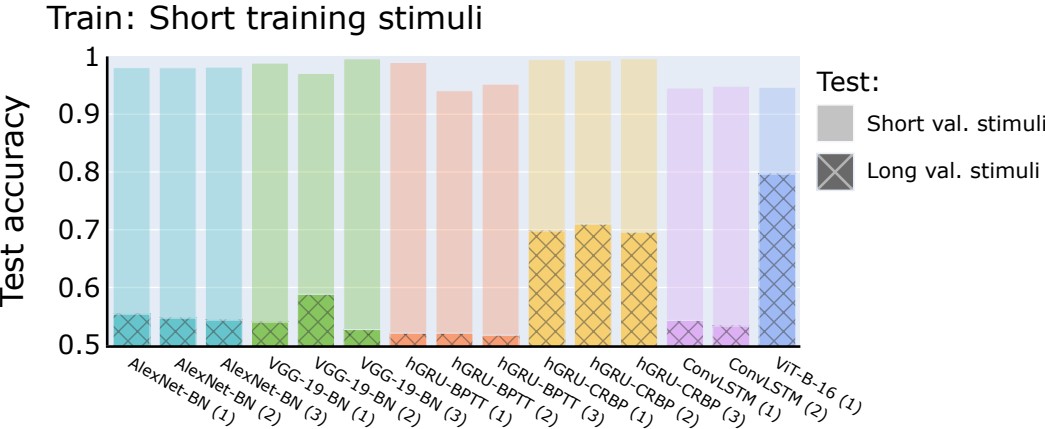

Figure S1: **Control models fail to generalize.** The graph summarizes the results from generalization experiments involving the incremental grouping task. To study generalization, all models were first trained on an easy version of the task with training stimuli for which the distance between the dots was short ($\leq 44$ px). They were then evaluated on validation stimuli with the same, short dot distances (transparent bars), as well as on validation stimuli with longer dot distances ($> 82$ px; cross hatched bars). The longer distances constituted a harder version of the task. hGRU-CRBP models, sharing the same architecture and contractor recurrent back-propagation (C-RBP) algorithm as the cRNN models studied in the main text, still performed well above chance level (0.50) on the hard version, attesting to their generalizability. The control models, in contrast, struggled with the hard version of the task, with ViT-B-16 being a noteworthy exception.

# C   Dynamically Expendable Computational Budget

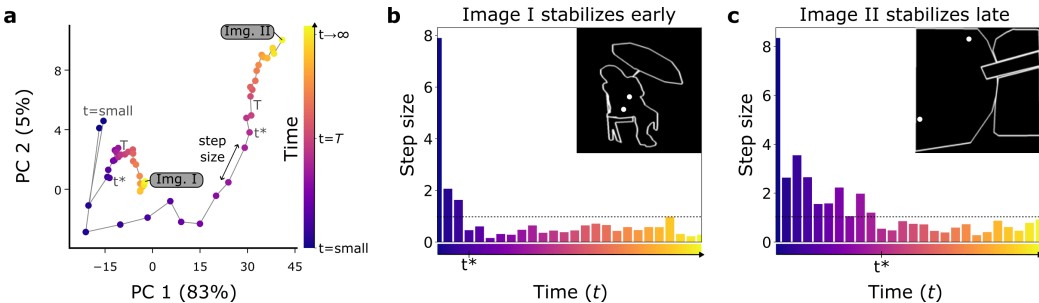

Figure S2: **Dynamic computational time. a.** Low-dimensional projections of our model dynamics for two input conditions in the incremental grouping task (Section 4 of the main text). For the easy input (Panel b), the model effectively "stops" computing at an earlier time point ($t^*$), as evidenced by dramatically reduced step sizes in the latent space. For the harder input (Panel c), step sizes remain large (above the dotted line) for a longer period. Overall this demonstrates that our model dynamically utilizes its compute budget in a stimulus-dependent manner. **b,c.** Alternative visualization of the step size as a function of time for the easy and hard stimuli, respectively. The dotted line here is arbitrarily chosen for visual comparison.

# D Videos

For the incremental grouping task (Section 4 in the main text) and the maze task (Section 6 in the main text), the cRNN learned a filling-in strategy that was evident from inspecting the latent activity $\mathbf{h_t}$ over time steps ($t$). The gradual spread of activity can be appreciated best in video-format. Examples of such videos can be found on our project page: https://serre-lab.github.io/rnn_rts_site.

Each video consists of three panels. The left panel shows the input to the model. The middle panel shows the uncertainty curve, with a marker added at every time step to indicate the progression of time. The right panel shows a visualization of the latent activity over time. Each frame shows $\mathbf{h_t}$ averaged over the channel dimension and smoothed by additionally averaging between $\mathbf{h_t}$ and $\mathbf{h_{t+1}}$. With each frame in the video, one can observe the spread of activity and the corresponding change in uncertainty.

# E    Distance Experiment

When humans perform the incremental grouping task (Section 4 in the main text), their RT tends to increase with increasing distances between the dots. This effect has been attributed to a serial spread of attentional resources [47, 52]. To test whether the effect was also present in the cRNN we trained, we created a dedicated test set, derived from a random subset of the validation outlines ($N = 150$). As explained in the main text, we repositioned the dots to be in the center of an object and 14 px apart in Euclidean distance. We then systemically moved them 14 px further apart until they hit an object boundary, thus creating different distance conditions for a given outline (see Fig. S3 for an example). This procedure resulted in a total of 1852 stimuli. Fig. S3 demonstrates how the distance manipulation affected the uncertainty curves for one example outline. The cRNN spent more time being uncertain if the dots were placed further apart, which is captured in our $\xi_{cRNN}$ metric. Analyzing the relation between distance and $\xi_{cRNN}$ across the whole test set using a linear mixed-effects model, we found a significant positive effect, $b = 1.46, SE = 0.11, z = 13.58, p < .001$ (Fig. S4).

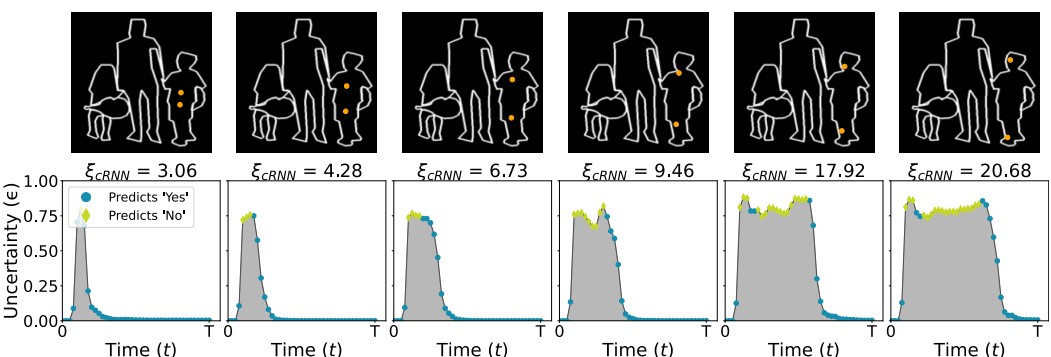

Figure S3: **Manipulating dot distance in the incremental grouping task.** The top row demonstrates the manipulation of the dot distance for an example outline. The dots are colored orange here for visualization purposes. The bottom row represents the corresponding uncertainty curves, with $\xi_{cRNN}$ annotated on top. With the dots further apart, the cRNN spends more time being uncertain, resulting in higher $\xi_{cRNN}$.

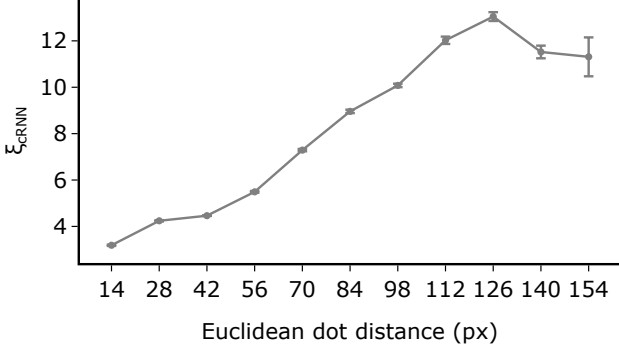

Figure S4: **Overall distance effect in the incremental grouping task.** The graph shows the average $\xi_{cRNN}$ across stimuli as we systematically moved the dots 14 px more apart. Error bars represent the standard error of the mean. The cRNN exhibits a clear distance effect, as did human RTs in [47].

# F   Non-Euclidean Distance Metric

Jeurissen et al. [47] describe a growth cone distance metric that is sensitive to the topology of the object, proximity of boundaries, and narrowness. The authors construct a series of non-overlapping, varying-sized discs whose centers trace a curve (originating at the fixation location and ending at the cue location). The radii are such that a disc is the maximum inscribed disc at its location. Disks never cross object boundaries. The final distance metric is defined as the total number of discs that constitute this curve

# G    Spatial Anisotropy in Incremental Grouping - Additional Experiments

As a further test of how factors beyond Euclidean distance might affect $\xi_{cRNN}$ in the incremental grouping task, we created two sets of stimuli (Fig. S7, Fig. S8), each derived from the same 23 novel outlines by placing dots in different configurations. Following the example of [47], each stimulus featured two adjacent objects of the same object class. In one set (Fig. S7), we created two conditions by placing one of the dots (or both) in a narrow region of an object in one condition, and moving the pair of dots to a wider region in the other condition. The Euclidean dot distance, as well as the topological distance were matched between both conditions. In the other stimulus set (Fig. S8), we used the same outlines but with different dot conditions. Here, one condition had the dots placed in a such a way that they could be connected by a straight path. In the other, the path was curved making that the higher topological distance condition. The Euclidean dot distance, on the other hand, was still matched. Fig. S5 visualizes the effect of both these manipulations on $\xi_{cRNN}$. As reported in the main text, paired t-tests showed that $\xi_{cRNN}$ was significantly higher in the narrow versus wide condition, $d = 0.70, t(22) = 3.34, p = .003$, and in the curved versus straight condition, $d = 0.98, t(22) = 4.71, p < .001$.

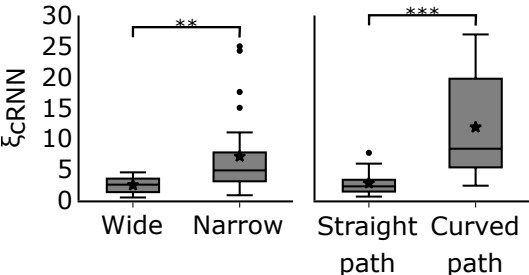

Figure S5: **Effects beyond those of Euclidean distance in the incremental grouping task.** We created two novel test sets (Fig. S7, Fig. S8). In one we manipulated whether the dots were placed in a wide versus narrow region of an object. In the other the path between the dots was either straight or curved, thereby manipulating topological distance. In both manipulations, Euclidean distance was kept constant. The boxplots show the effect of these manipulations on $\xi_{cRNN}$. The star markers inside the boxes represent the condition mean. The asterisks on top represent the $p$-values resulting from a paired t-test, $** < .01, *** < .001$.

Finally, in a last follow-up experiment, we varied the distance between one dot and an object boundary while keeping the Euclidean distance between the dots themselves constant (Fig. S6). We found that our model exhibits higher values near the boundaries (Fig. S6b,c). This is strikingly similar to human behavior presented in prior literature ([47], Fig. 3d).

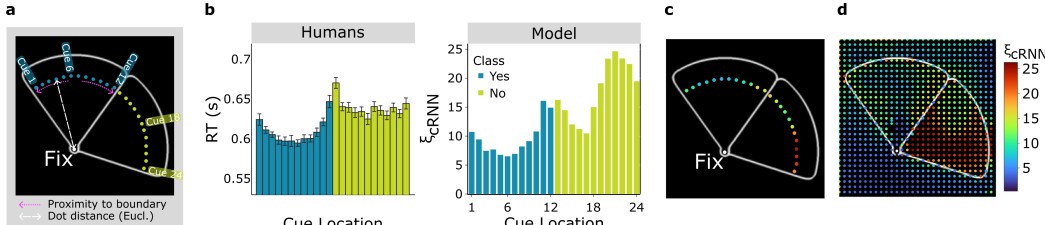

Figure S6: **Fixed Euclidean distance; varying distance to boundaries. a.** Wedge stimuli presented to our model. **b.** We observe higher $\xi_{cRNN}$ closer to boundaries, as was observed for human RT in [47]. Data from [47] are shown on the left here. **c.** Visualizing our model RT predictions for a fixed chosen radius **d.** The complete SU-map reveals a further disparity between the "yes" and "no" conditions.

Narrow    Wide    Narrow    Wide    Narrow    Wide

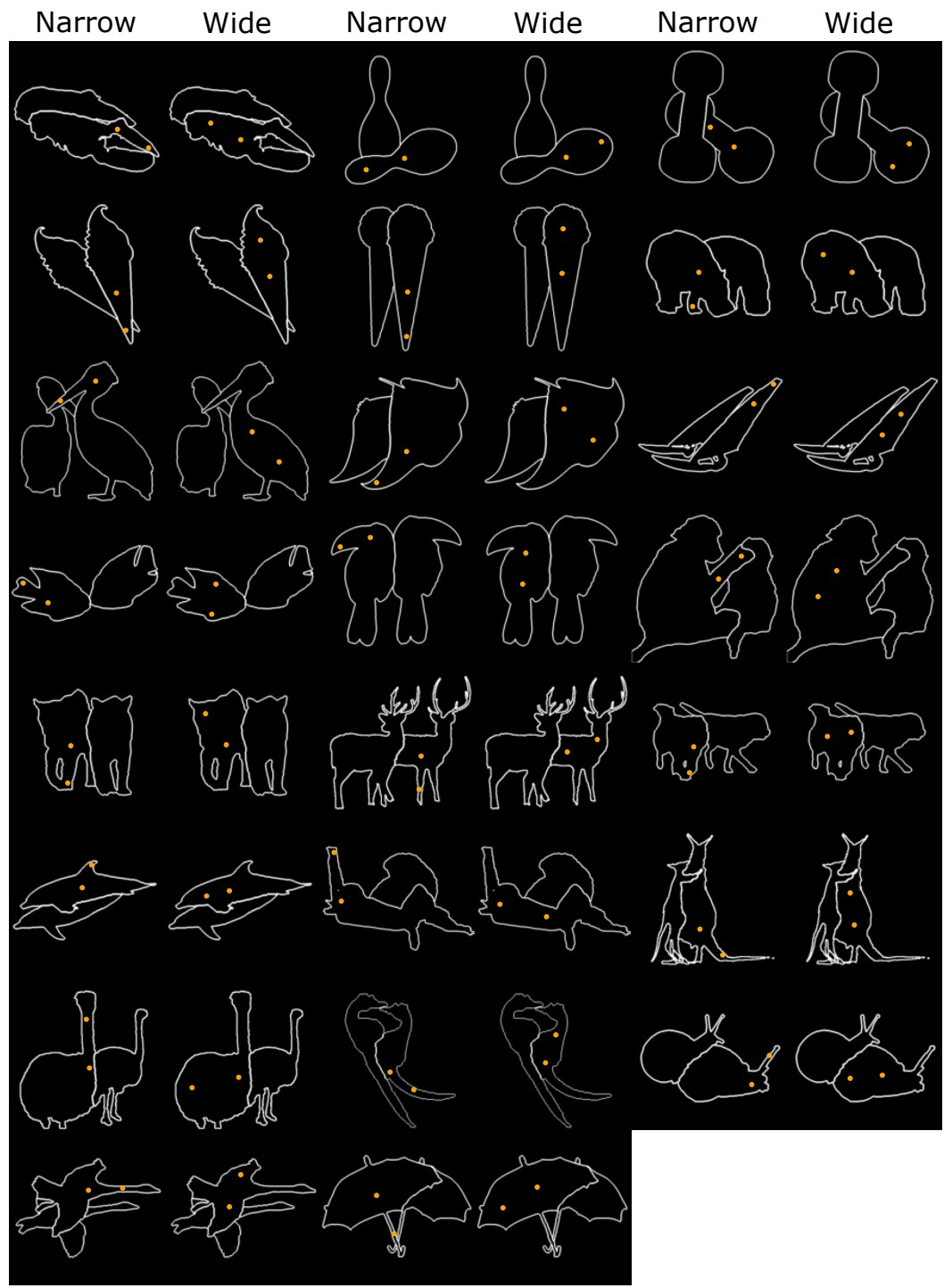

Figure S7: **Novel outline stimuli for the incremental grouping task featuring a narrow versus wide manipulation.** Dots are colored orange for visualization purposes.

Curved    Straight    Curved    Straight    Curved    Straight

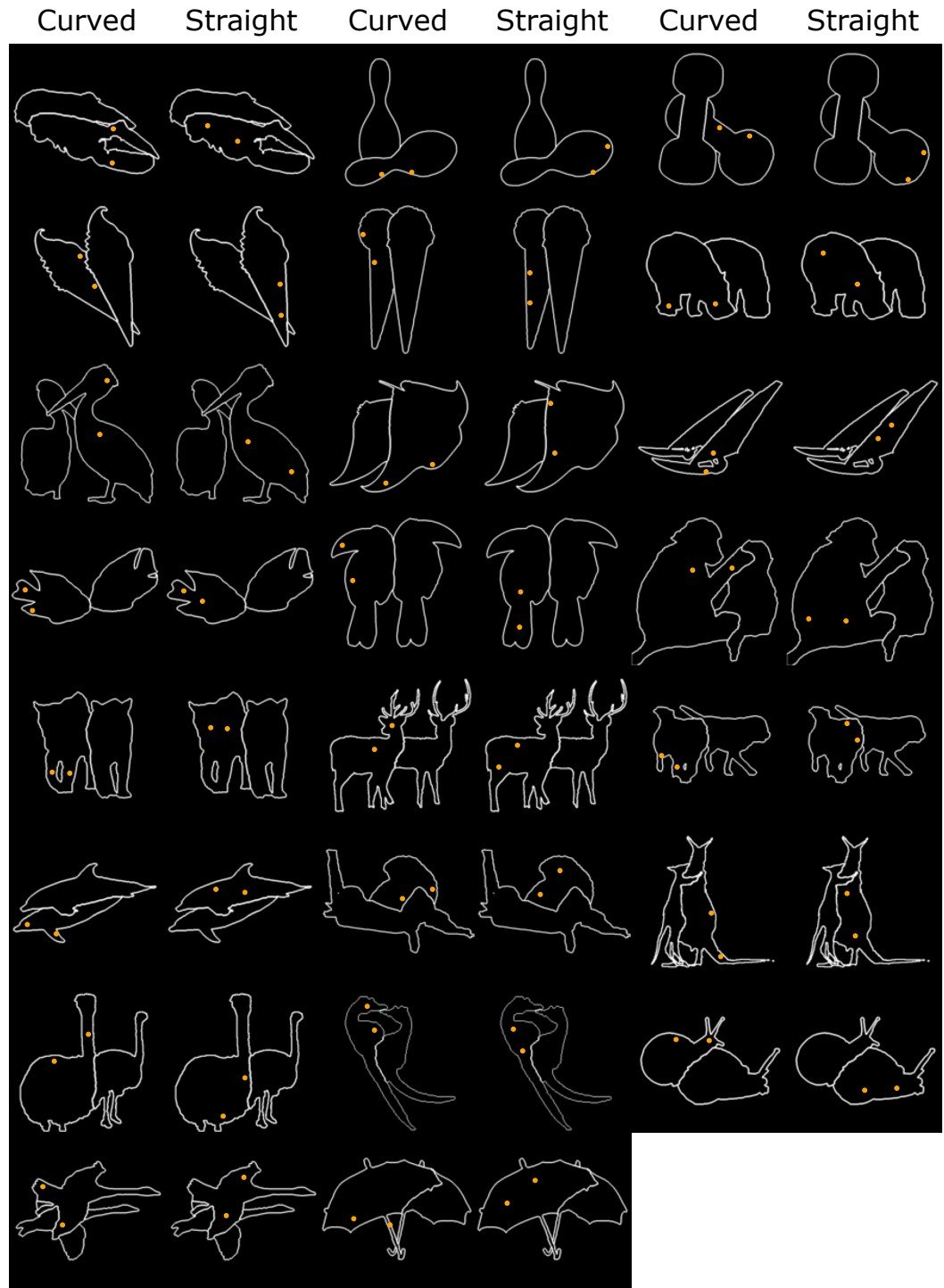

Figure S8: **Novel outline stimuli for the incremental grouping task featuring a topological distance manipulation.** The dots were placed such that it was either a straight or a curved path (higher topological distance) connecting them, whereas the Euclidean distance was the same. Dots are colored orange for visualization purposes.

## H  Ambiguous Grouping Stimuli and Non-zero Uncertainties

Here, we take a closer look at which stimuli elicit very high $\xi_{cRNN}$ values. Looking at our validation set first, we observed that some high-$\xi_{cRNN}$ stimuli are characterized by uncertainty curves that remain high, even at $t = T$, suggesting the model deems them ambiguous (see Fig. S9 for an example).

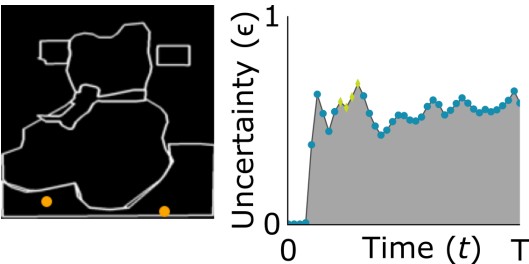

Figure S9: **High uncertainty example.** Some challenging samples in our validation set display non-zero uncertainty.

Furthermore, in a test set we designed to be ambiguous (it features occlusions), we frequently observed such non-decreasing uncertainty curves too (see Fig. S10). We note that while our training procedure promotes stable attractor states in the model's *latent* dynamics, this is disparate from readout dynamics allowing the framework to truly explore the spectrum of computation times and uncertainties.

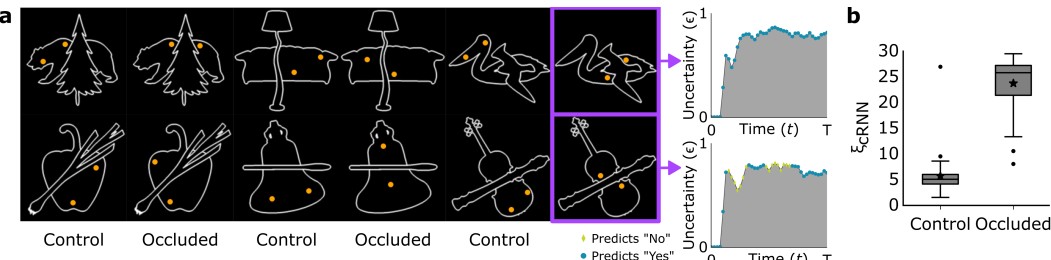

Figure S10: **Testing model uncertainty on occluded stimuli. a.** We created a dataset (N=76) of occluded images, half of which have both dots on the same side of the occluding object (control). The Euclidean dot distance is matched between the control and occluded condition. We performed zero-shot transfer with our trained cRNN on this set and observed that for the occluded samples, model uncertainty often remained at high (non-zero) values in the time limit. **b.** A paired t-test reveals that our metric is significantly higher in the occluded condition compared to the control condition: Cohen's d = 2.85, t(37) = 17.54, p < .001.

# I Additional Maze Analyses

Human participants have been found to take more time to solve a maze when the solution path features more junctions [56]. To study whether the cRNN we trained exhibited a similar slow-down, we computed two measures for the Yes-mazes (Fig. S11). One was the number of T-junctions encountered on the shortest path between the two cues. The other was the total number of T-junctions on the maze segment containing the cues. Linear regression analyses that statistically controlled for path length by including it as a predictor, did not yield evidence for an effect of the first measure, $b = -0.23, SE = 0.29, t(234) = -0.79, p = .43$ or the latter, $b = 0.28, SE = 0.18, t(234) = 1.56, p = .12$. We treat this finding as an important example for how our metric can help identify potential mismatches between human- and model-behavior and ultimately guide the development of better aligned models.

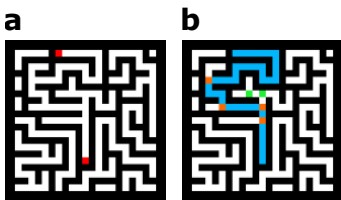

Figure S11: **Counting T-junctions in Yes-mazes. a.** An example maze. **b.** Visualization of the T-junctions. The solution path connecting the cued locations for the same maze shown in the previous panel is indicated in blue here. We found no evidence for an effect of the number of T-junctions included in the solution path (orange squares), nor of the number of T-junctions included in entire maze segment containing the cues (orange + green squares) on $\xi_{cRNN}$.

# J ConvLSTM Results on Scene Categorization

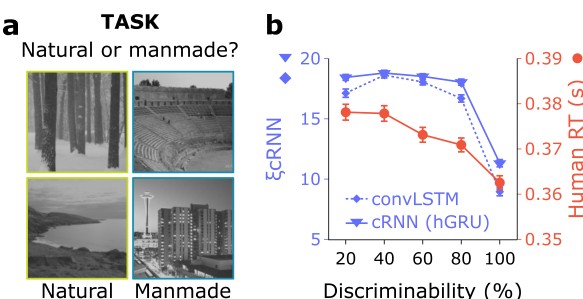

Figure S12: **A convLSTM equipped with our metric captures human RT patterns in a scene categorization task.** Within our framework, we are able to swap out various model (or architecture) choices without affecting the downstream formulation. Whereas our choice of architecture in the main text was motivated by the demands that a task like incremental grouping imposes, we show here that $\xi$ can also be computed for other architectures, such as a convLSTM. The fact that our metric is model-agnostic makes it ideally suited for comparing and contrasting models in terms of their temporal alignment.

