# OpenReview forum: "Computing a human-like reaction time metric from stable recurrent vision models"
_NeurIPS.cc/2023/Conference — NeurIPS 2023 spotlight_

### Official Review · Reviewer_4B3C · 2023-06-15

**Soundness:** 3 good
**Presentation:** 3 good
**Contribution:** 3 good
**Rating:** 6
**Confidence:** 4

**Summary:**

This paper introduces a new measure of visual artificial network computation 'time', $\xi_{cRNN}$, as the time-averaged uncertainty of a convolutional RNN trained with an evidential deep learning loss (Sensoy et al.). The authors proceed to analyse the dynamics of this network as it solves a range of classification tasks. Importantly, beyond the network learning to solve the tasks, $\xi_{cRNN}$ is generally well correlated with human reaction times, with which it also shares several qualitative features across tasks.

**Strengths:**

Deep network models of the visual system have often focused purely on classification accuracy and neural representations when comparing to biological data. However, this neglects an important aspect of human behaviour, namely the time it takes to arrive at a decision. The present paper tackles this important question using recent ideas from the machine learning literature and represents an interesting step in the direction of capturing the temporal variability in human behaviour.
The authors analyse an impressive breadth of tasks and behavioural data and include many interesting analyses, both qualitative and quantitative.

**Weaknesses:**

The major weakness of the submission is that while there are strong _correlations_ between $\xi_{cRNN}$ and human reaction times, there is less evidence that $\xi_{cRNN}$ is mechanistically similar to human reaction times, since it is fundamentally a measure of uncertainty rather than computation time. In particular, human reaction times generally involve a tradeoff between computation/evidence accumulation and decision making (as in common drift diffusion models). On the contrary, the cRNN has a fixed computational budget and has no need or even capacity for evaluating this tradeoff. This is in contrast to a few previous deep learning models in the literature that are capable of explicitly trading off computation and actions (e.g. Graves et al., Pascanu et al., and refs by the authors in the ML literature, and Jensen et al. in the Neuro/Cogsci literature). These considerations are important because e.g. task difficulty is likely to correlate with both uncertainty and reaction time, and this raises the question of whether a model of one is automatically a model of the other.

It might the interesting to compare the current model to alternative models that more explicitly have adaptive/variable computation time, such as a cRNN that computes until $\epsilon$ reaches a certain threshold (akin to classical drift diffusion models).

As the authors mention in L326, another potential weakness is that the present approach does not easily generalize beyond classification tasks, which form a small subset of the types of problems humans and animals are faced with during natural behaviour.

_References:_ \
Graves et al.: "Adaptive computation time for recurrent neural networks", arXiv 2016.\
Pascanu et al.: "Learning model-based planning from scratch", arXiv 2017.\
Jensen et al.: "A recurrent network model of planning explains hippocampal replay and human behavior", bioRxiv 2023.\
Bansal et al.: "End-to-end Algorithm Synthesis with Recurrent Networks: Extrapolation without Overthinking", NeurIPS 2022.

**Questions:**

L106: what does 'without loss of generality' mean here?

L123 "our control models lack inference-time flexibility." What does this mean?

L144: It might be worth comparing to or discussing recent work by Bansal et al. (NeurIPS 2022), which develops a way of stably training convolutional RNNs with long computation times that generalize from small to large problem settings.

L156 "for a complete mathematical specification of this loss, refer to SI 1". $\epsilon$ does not seem to be defined in either the main text or SI beyond a qualitative description as 'uncertainty'. It would be good to provide a precise mathematical definition somewhere.

L204: it would be good to briefly explain how this distance measure by Jeurissen et al. is defined (or at least refer to SI where it can be described in more detail).

Figure 3b and 5c: it's not entirely clear to me how these 'activity maps' are computed?

Figure 7b: it might be worth talking a bit more about the differences between the human and model data here instead of doing linear regression and calling it a day. The cRNN data is decidedly not linear and exhibits something resembling a plateau followed by a sharp drop at high discriminability. This seems rather different from the monotonic decrease in human RT, and the authors could perhaps speculate on or discuss this difference.

Supplementary material: it would be worth having a separate folder with just a few example videos of each condition in addition to the larger set of videos currently provided.

**Limitations:**

The authors have adequately discussed the limitations of the work.

---

> ### Author Rebuttal · Authors · 2023-08-10
>
> We thank the reviewer for their positive feedback and useful comments. We offer clarifications, additional numerical analyses, and visualizations to address concerns raised in this review. We hope the reviewer sees value and effort in our responses and is willing to adjust their score.
>
> > **Fixed computational budget of the cRNN**
>
> Thanks for allowing us to clarify this point. We draw an important distinction between a model having a fixed "total" budget versus a budget that can be expended *dynamically* in a stimulus-dependent manner. Consider the results presented in **Fig. R5**. Inspecting the step sizes of the recurrent dynamics (in a low-dimensional latent space) reveals that the cRNN utilizes its computational budget differently for  easy (**Fig. R5b**) versus hard (**Fig. R5c**) stimuli. The network practically *stops* (i.e., step size becomes small) after $t=t^*$, where $t^*$ is stimulus-dependent. This is unlike standard BPTT-trained cRNNs and shows how our cRNN computes dynamically and has the potential to evaluate the tradeoff between computation/evidence accumulation.
>
> > **Alternative models with explicit adaptive computational time**
>
> We appreciate this suggestion. We now implement and include comparisons to cRNNs trained with ACT on the incremental grouping task. Here, we report quantitative results for one out of three well-performing (val. acc. $\geq 0.95$)  runs. For the other two, we note that the results were qualitatively similar. None of these models showed a significant *positive* correlation between the ACT step count ($\rho$) and human RT, $r= -.26, p=.04$. While for some (but not all), regressing $\rho$ onto Euclidean distance yielded a significant positive linear slope, $b=0.07$, $SE=0.02$, $z=4.20$, $p<.001$, none of the models seemed to capture the effects of other factors known to affect human RT: narrow vs. wide, $d=0.38$, $t(21)=-1.82$, $p=.08$; curved vs. straight, $d=0.03$, $t(21)=-0.13$, $p=.90$; and A vs. B vs. C vs. D in **Fig. 2c**, $F(3,63)=1.84$, $p=.15$.
>
> We hypothesize that despite ACT being a clever scheme for small networks, the fundamental bottleneck arises from the memory demands imposed by BPTT that limit step counts for cRNNs condemning any derived time metric to be coarse. We also note that these are only preliminary results and warrant further exploration.
>
> > **Beyond classification tasks**
>
> We agree that this is a very promising direction for future research. Alternative forced choice tests comprise a dominant part of visual cognitive psychology paradigms for which researchers have amassed a wealth of human RT data. And thus, our motivation to start here. To the best of our knowledge, we present one of the early attempts to leverage RT metrics for alignment, and we are confident in expanding the scope of this work in the future.
>
> > **L106: what does 'without loss of generality' mean here?**
>
> We have clarified this as follows:
>
> The model of interest chosen in this work is the horizontal gated recurrent unit (hGRU; [19]), a convolutional recurrent neural network model that we will canonically refer to as cRNN throughout the rest of the paper. This choice does not imply any loss of generality, as our framework applies to any cRNN architecture.
>
> > **L123 "our control models lack inference-time flexibility." What does this mean?**
>
> By the *lack of inference-time flexibility*, we refer to the inability of our control models trained on easier task settings to generalize to arbitrarily harder task settings.
>
> > **Comparison to Bansal et al. (2022)**
>
> Thanks for this reference. We will factor this into our revised discussion.
>
> > **Mathematical definition of $\epsilon$**
>
> Agreed! We have added this in the description.
>
> Starting at **L41** in **S1.2**: “$S = \sum_{j=1}^{K}\alpha_j$ represents the "total evidence" accumulated by the model (or the Dirichlet strength). $D_{KL}$ is the Kullback–Leibler divergence measure. $\boldsymbol{\hat{\alpha}}$ are the Dirichlet parameters of just the "misleading" evidence given by $\boldsymbol{\hat{\alpha}} = \mathbf{y} + (1-\mathbf{y})\boldsymbol{\alpha}$. We define the instantaneous uncertainty as $\epsilon = \frac{K}{S}$.
>
> > **Distance measure from Jeurissen et al.**
>
> We will add this description to the SI. "Jeurissen et al. [41] describe a growth cone distance metric that is sensitive to the topology of the object, proximity of boundaries, and narrowness. The authors construct a series of non-overlapping, varying-sized discs whose centers trace a curve (originating at the fixation location and ending at the cue location). The radii are such that a disc is the maximum inscribed disc at its location. Disks never cross object boundaries. The final distance metric is defined as the total number of discs that constitute this curve."
>
> > **Computing the activity maps**
>
> Thanks for the comment. We mention this in the main text (**L174**). We will repeat it in the figure caption to make it easier for the reader.
>
> "First, we track cRNN dynamics by looking at its latent activity $\mathbf{h_t}$ at every time step $t$, averaged across the channel dimension and smoothed by averaging $\mathbf{h_t}$ and $\mathbf{h_{t+1}}$."
>
> > **Clarification on Figure 7b**
>
> Point well taken! We apply our cRNN on this scene categorization task to showcase the generality of our framework in extending to naturalistic stimuli. While the ability of our metric+cRNN to capture aspects of variations in scene discriminability is a promising sign, we acknowledge that there could be other cRNN architectures that are better suited for this particular task and are subsequently better aligned. We intended to propose our metric as a tool to precisely enable these sorts of comparisons.
>
> > **Re-organizing SI videos**
>
> Thanks for this suggestion. We will keep your comment in mind to highlight representative examples (per-condition) in our project webpage (URL currently protected for double-blindness).

---

> > ### Comment · Reviewer_4B3C · 2023-08-10
> >
> > I appreciate the detailed response of the authors to myself and the other reviewers.
> >
> > ### Fixed computational budget
> >
> > I appreciate that network iterations might change the latent space less at later iterations for the simpler task compared to the more complex task, and that this intuitively seems to suggest that the model should be able to rely on less computation in this case. However, at the end of the day, the two settings use the same number of network iterations and therefore FLOPs (unless I've misunderstood something). It is exactly this intuition that was the motivation for suggesting an 'adaptive computation time' network, and it is interesting that this did not work since it seems like computation could be terminated earlier for simpler tasks, and the results of the authors show that the network has lower uncertainty earlier for simpler tasks. I wonder if there is another criterion that would be more suitable for 'early stopping' of the computation to develop a network that truly has a 'reaction time' rather than only an uncertainty?
> >
> > ### Without loss of generality
> >
> > I would suggest not saying "without loss of generality" and just mentioning that the framework generalizes to other cRNN architectures.
> >
> > ### Figure 7b
> >
> > I appreciate that your framework of course cannot explain all empirical results, and that being able to identify results that are/are not consistent with a given framework is part of its value. However, for this reason, it's also important to be up front about the cases where the data differs from the model.

---

> > > ### Author Response · Authors · 2023-08-11
> > > **A productive discussion on capturing uncertainty vs. time**
> > >
> > > Thanks for reading our rebuttal and your prompt response.
> > >
> > > > Computational budget
> > >
> > > Another insightful comment, and we sympathize with their FLOPs remark. Typically, techniques for early stopping involve at least one (or more) heuristic choices, such as the threshold value in ACT, that need careful finessing. We try to build a formulation that does not enforce such a selection. By training our cRNN with an attractor dynamics constraint, we wanted to achieve stable latent states that would also result in stable readouts (and thus a stable, bounded metric).
> > >
> > > One observation we like to highlight is that achieving lower uncertainty states does not immediately suggest that these states were reached quickly. In other words, models in our framework can achieve low uncertainty states eventually after a prolonged period of high uncertainty. Naturally, there are cases where the model reaches attractor states with low uncertainty rapidly (for very easy stimuli), or gradually stabilizes in states with high uncertainty. So in essence, our metric also implicitly captures aspects of time since it integrates over instantaneous uncertainty. Empirically, we also find this to be critical. This integrated metric, as opposed to just the uncertainty value at the final state, consistently aligns better with human RTs.
> > >
> > > Thinking out loud, one way to make these two (no heuristics *and* explicit measure of time) desires meet might be to learn a controller policy (with RL) that can choose "stop" as an action. Though admittedly, this is a more challenging optimization problem. We will plan to add this comment to our extended discussion section.
> > >
> > > > Without loss of generality
> > >
> > > Thanks. We will follow your suggestion here.
> > >
> > > “*The model of interest chosen in this work is the horizontal gated recurrent unit (hGRU; [19]), a convolutional recurrent neural network model that we will canonically refer to as cRNN throughout the rest of the paper. In principle, our framework applies to other cRNN architectures as well.*”
> > >
> > > > Figure 7b
> > >
> > > Fair enough! We will include this text in manuscript Section 7.2 (Scene categorization; Results):
> > >
> > > *While our cRNN models of choice here do not perfectly account for empirical patterns in human reaction times, specifically in their non-monotonic reaction time readouts for the low-discriminability stimuli, they provide a promising start.*
> > >
> > > And this text in manuscript Section 8 (Discussion):
> > >
> > > *The value proposition of this framework includes the ability to adjudicate models based on the consistency of their latent dynamics with human reaction time data.*

---

> > > > ### Comment · Reviewer_4B3C · 2023-08-15
> > > >
> > > > Thanks a lot for the additional response!
> > > >
> > > > I agree that any handcrafted 'early stopping' criterion is bound to be heuristic so makes sense re. the attractor constraints.
> > > >
> > > > I also understand why the integrated uncertainty is mostly better than the final uncertainty - although if the system does reach an attractor state, the two would be equivalent if you took the total allowed computation time to infinity. It is also in some ways at odds with biological reaction times, where a fast RT can't depend on what the agent's uncertainty would be at some point in the future.
> > > >
> > > > Using RL to train agents that can choose when to stop would definitely be an interesting extension, although of course well beyond the scope of the present paper. The Pascanu et al., and Jensen et al. references in my original review do something along these lines for more 'canonical' RL tasks, and it would be interesting to see how this approach would compare to human behaviour for a perceptual decision making task.

---

### Official Review · Reviewer_fhSL · 2023-07-06

**Soundness:** 4 excellent
**Presentation:** 4 excellent
**Contribution:** 3 good
**Rating:** 8
**Confidence:** 4

**Summary:**

The authors in this work propose a combination of model output uncertainty predictions along with stable recurrent vision architectures in order to derive a proxy for models' reaction time to process input static images. The authors use the previously published horizontal GRU architecture combined with stable RNN training methods (contractive RBP) in their experiments. The proposed work is technically sound with extensive evaluation on 4 diverse datasets (some inspired by prior work in visual psychology) where the computed proxy model reaction time ($\xi$) trends correlate positively with that of humans performing the same task. Overall, this paper contributes a new way to estimate output uncertainty of recurrent convolutional networks and to evaluate similarity between model & human perception by taking into account temporal dynamics of processing static inputs.

**Strengths:**

+ The authors make several key contributions: (1) combining Evidential Deep Learning-based output uncertainty prediction with convolutional RNNs and using the AUC of the output uncertainties over time as a measure of the model's reaction time; (2) training stable recurrent vision models using the above EDL-based readout + objective function in order to obtain models whose temporal dynamics of processing visual inputs matches that of humans performing the same task.
+ The experiments performed are thorough on all datasets, it is clear that the positive correlation between model and human reaction times is present on the tasks evaluated.
+ If the authors were to release the code for some of these datasets (especially the incremental grouping task in Fig 2), this would make the contribution even more valuable as it would encourage further exploration of these understudied grouping problems
+ Clear presentation; The authors have exactly stated their contributions and have written the paper with good clarity and detail. The figures are intuitive and appreciate the video presentations showing model activations and uncertainty through time in the Supplementary.

**Weaknesses:**

- Choice of tasks: While the presented 4 visual cognition tasks are interesting and relevant to cognitive scientists, only the scene recognition task concerns real-world stimuli and natural vision. It would be great if the authors found any interesting patterns of reaction time on natural images or more naturalistic tasks than the ones shown here.
- Advantages of similarity in reaction time: It is very interesting that the authors are presenting recurrent networks that are correlated with humans in terms of reaction time (and the benefits as a model of biological visual processing are clear), but does this similarity translate to any significant advantages for machine learning? Are models with better RT correlation with humans also well calibrated, or more robust compared to others? In summary, inter-model comparison based on how well they rank on RT similarity with humans would be an interesting direction that the authors don't comment about here.

**Questions:**

Please refer to my review above for questions.

**Limitations:**

The authors have adequately addressed limitations of this work.

---

> ### Author Rebuttal · Authors · 2023-08-10
>
> We thank the reviewer for their extremely positive feedback and very helpful comments. We have provided clarifications and additional numerical analyses below to address concerns raised in this review.
>
> > **The figures are intuitive and appreciate the video presentations**
>
> We really appreciate the reviewer for going through the supplementary information in detail and for watching the videos.
>
> > **Code and data sharing**
>
> We plan to make all code (model training/analyses) and datasets (with the accompanying data generation code) publicly available for the community to build on.
>
> > **Extensions to more naturalistic tasks**
>
> We agree! This is a great direction and one that we envision for this work in the future. In this manuscript, we curtail our focus to detailing a framework without over-engineering very large-scale models that typically are needed to be performant on naturalistic benchmarks. However, our framework does not preclude scalability. We look forward to accomplishing this in future work.
>
> > **Are there benefits for ML conferred by temporal alignment?**
>
> This is an excellent point! In the past, aligning deep neural network models to neuroanatomy, neural, and behavioral data has shown benefits in terms of performance (R1), interpretability (R2), and adversarial robustness (R3). One of our motivations for developing this framework was to enable investigations into the benefits of temporal alignment.
>
> While this was not a primary focus at the time of writing this manuscript, intrigued by the reviewer's comments we piloted experiments to test what we think is an important computational benefit conferred by temporal alignment: *generalization ability*.
>
> We consider two model classes (A and B) in the context of our incremental grouping task. cRNNs in Model Class A were *aligned* models that learned to solve the task. cRNNs in Model Class B (15x15 kernels, BPTT, 6 timesteps) also learned to solve the task (val. accuracy $\geq 0.99$) but were *not aligned*. They did not capture narrow vs. wide or curved vs. straight effects. They also did not correlate significantly with human RT: $r =  -.08,  p  = .51$. Interestingly, when we tested both these model classes for strong generalization (experimental details in **SI 2**), the aligned model class showed superior average performance (**0.70** vs **0.61**).
>
> Though this result is anecdotal and needs a more thorough investigation, we believe that there are benefits for ML that accompany model temporal alignment.
>
> [R1] Fel et al. (2022) Harmonizing the object recognition strategies of deep neural networks with humans.
>
> [R2] Kubilius, Schrimpf et al. (2019) Brain-Like Object Recognition with High-Performing Shallow Recurrent ANNs.
>
> [R3] Dapello et al. (2020) Simulating a primary visual cortex at the front of CNNs improves robustness to image perturbation.

---

### Official Review · Reviewer_JuhY · 2023-07-10

**Soundness:** 3 good
**Presentation:** 3 good
**Contribution:** 3 good
**Rating:** 7
**Confidence:** 5

**Summary:**

The authors of the paper present a novel concept of crafting a metric for analyzing the temporal alignment between Recurrent Neural Networks (RNNs) and human behavior. The innovative approach, rooted in the estimate of task uncertainty via the application of the Derelict distribution, is both interested and pertinent. In an era where understanding the interface between machine learning models, specifically RNNs, and human behavior is of utmost importance, the authors' research is timely and bears the potential for broad impact. A commendable aspect of this study is the diverse range of tasks that the authors have attempted to cover. Their inclusive approach elevates the practical relevance of their research, thus ensuring the applicability of their findings across multiple contexts. However, there are two crucial areas that the authors need to address to improve the overall quality and comprehension of the manuscript. Firstly, in the same/different dot object task, the authors discuss a correlation between the task's difficulty, reaction time, and dot distance. It is important to note that human reaction time also ties strongly with factors such as the object's topology and occlusion. The authors assert that their network and measurements reflect human reaction time concerning these more intricate features. Nevertheless, the demonstration of this claim in the manuscript is not lucid enough. To solidify their claim, it is crucial for the authors to condition the different stimulus conditions on the distance between dots. They should explicitly illustrate how elements like the narrowness of the object outline and occlusion correlate with human reaction time and their RNN metric. Secondly, the specificity of the proposed metric to the particular cortically inspired architecture employed in this research should be clarified. It would be beneficial to understand if this metric can be generalized to other architectures. For instance, would it be feasible to use this metric if one were to implement a convolutional LSTM or another vanilla RNN? The authors' response to this concern could significantly influence the wider application of their proposed metric.  In conclusion, while the paper is insightful and tackles a highly relevant topic, it is recommended that the authors address the concerns raised to improve the robustness and clarity of their research. By doing so, the potential impact of the manuscript can be further augmented, providing a more substantial contribution to the field.

**Strengths:**

A relevant and timely paper extending the comparison between human behavior and ANNs.

**Weaknesses:**

Not clear how well they demonstrate that their metric predicts human reaction time between the more straightforward relationship of dots distance, task difficulty and reaction time. In other words do they predict reaction time and object occlusion, and more complex topology of objects (narrowness of segmentation boundaries etc).

Does their metric extend to other architectures beyond the specific one they use here.

**Questions:**

Perform analysis to condition on dot distance to demonstrate that other more intricate relationships predict human reaction time.

**Limitations:**

Yes

---

> ### Author Rebuttal · Authors · 2023-08-10
>
> We thank the reviewer for their positive feedback and very helpful comments. We have provided clarifications and additional numerical analyses below to address concerns raised in this review.
>
> > **Moving beyond Euclidean distance between cue and fixation dots to other factors such as object topology**
>
> Thanks for this comment. We agree with the reviewer that human reaction times are affected by factors beyond the Euclidean distance between cue and fixation dots in our incremental grouping task. We perform several analyses in the manuscript (and further ones in this rebuttal) to back this claim. We will take this opportunity to highlight these here and better position them in the main text.
>
> #### Narrowness and topological factors
> We explore stimulus manipulations presented in Ref [41] in detail in this manuscript (subsection starting at L198 and titled “$\xi_{cRNN}$ recapitulates the spatial anisotropy in human RT data”). We consider four distinct manipulations (A-D) in the incremental grouping task. The cue and fixation dot are nearest in Condition A. Condition B and Condition C are *matched* in Euclidean distance (larger than that in Condition A) but Condition C has the cue dot on *narrow* parts of the image. While Conditions A, B, and C correspond to the cue and fixation dot on the same object, Condition D has counterbalanced cue locations for the "different" case.
>
> 1. We find that $\xi_{cRNN}$ is significantly higher in Condition C than Condition B even though they are matched in Euclidean distance, consistent with prior work on human RTs. This suggests that our metric clearly captures non-Euclidean factors (**Fig. 2c; L212** in the manuscript).
>
> 1. The variance in $\xi_{cRNN}$ is significantly better explained by a non-Euclidean distance metric (introduced in [46], explanation provided in our response to **4B3C**). We report this on **L208**.
>
> 1. To further test effects beyond Euclidean distance we generated a novel stimulus set that explores two additional manipulations. In the first test, we keep *both* Euclidean distance and topological distance constant, but manipulated whether the path between the dots passed through narrow regions. $\xi_{cRNN}$ was significantly higher in the Narrow condition than in the Wide condition. In the second test, we kept the Euclidean distance constant but manipulated whether there was a straight path between the dots or not (making the latter condition higher in topological distance). $\xi_{cRNN}$ was significantly higher in the latter condition. **These results are summarized starting from L216 in the main text and discussed in more detail in the supplementary information. Fig. S5 and Fig. S6 of the SI also show all stimuli used.**
>
> #### Fixed Euclidean distance. Varying distances to boundaries.
>
> In a follow-up experiment, we varied the distance between the fixation dot and an object boundary while keeping the Euclidean distance between the cue and fixation dot constant (**Fig. R4a**). We found that our model exhibits higher $\xi_{cRNN}$ values near the boundaries (**Fig. R4b,c**). This is strikingly similar to human behavior presented in prior literature (Ref [41], Fig. 3d).
>
> #### Occlusions
>
> As we discuss in our response to reviewer **yVHQ**, we also test our model on "occluded" stimuli (**Fig. R1**). When compared to control conditions, we find that our model exhibits extended periods of higher uncertainty in the occluded stimuli (**Fig. R1b**).
>
> > **Specificity of the proposed metric to the architecture considered in the manuscript**
>
> This question motivated us to clarify an aspect of our framework and we have now taken steps to perform additional model comparisons. We believe this makes our manuscript stronger and we thank the reviewer for that.
>
> First, we take this opportunity to clarify that, mathematically, the choice of cRNN architecture and our evidential metric can be disentangled. Suppose $f_\theta$ is our cRNN with parameters $\theta$ specifying the model architecture. For the evidential loss term, we rely on interpreting the outputs of a readout function $g(.)$ operating on the cRNN states in this following manner: $\hat{\boldsymbol{\alpha}}(t) = g(f_{\theta}^t(x))$. Here $x$ denotes the stimulus and $f^t$ denotes the recursive application of function $f$, $t$ times. The subsequent construction of the evidential loss as well as our metric rely only on $\hat{\boldsymbol{\alpha}}(t)$. Within this framework, we are able to swap out various choices of $\theta$ and $f$(model architecture) without affecting the downstream formulation. This makes our metric an ideal choice to compare and contrast models in terms of their temporal alignment.
>
> Having said that, our choice of architecture was motivated by the demands that a task like incremental grouping imposes. The visual cognitive psychology literature posits that incremental grouping requires feedback instantiated by lateral connectivity. Empirically, we find this as well. A recurrent CNN without horizontal connectivity, such as a convolutional LSTM (convLSTM), fails to learn generalizable solutions on this task (**Fig. S1**) and hence was not an appropriate model choice to test our metric.
>
> However, the rapid scene categorization task presents an opportunity to demonstrate the generality of our metric. This task is more akin to purely perceptual processing and relies less heavily on lateral feedback. We implement and train a convLSTM equipped with our metric on this task and indeed find that our metric captures variations in scene discriminability (**Fig. R3**), $b = -0.48,SE = 0.03,t = -16.88,p < .001. $ The correlation between $\xi_{convLSTM}$ and human RT is $.17, p<.001$.

---

### Official Review · Reviewer_yVHQ · 2023-07-14

**Soundness:** 3 good
**Presentation:** 4 excellent
**Contribution:** 3 good
**Rating:** 8
**Confidence:** 3

**Summary:**

This study introduces a metric for evaluating the alignment between model and human behavior wrt task complexity reflected in reaction times. The metric is easy to compute and shows qualitative correspondence with human RTs in different tasks.

**Strengths:**

The paper is well written and easy to understand.

The study attempts to go beyond choice responses to establish correspondence between model and behavior. For image-computable models, this is both novel and important.

The idea of using model uncertainty as a proxy for RT is intuitive and appealing.

**Weaknesses:**

The models are trained using an algorithm that imposes attractor dynamics which causes instantaneous  uncertainty to become zero. This is critical for their proposed metric to remain bounded. But this is a bit odd because subjective uncertainty should be non-zero for difficult tasks even if given a long time. This raises the question of whether the metric is meaningful for other tasks e.g. with ambiguous stimuli.

The authors make a convincing case for the utility of this metric for evaluating models. But the metric itself is artificial and might not have a correlate in neural activity. An alternative (and more brain-like) but admittedly more challenging approach would be to  learn a policy that chooses left/right/wait at each moment (e.g. via RL) based on the latent representation such that you can directly get a reaction time from the model. This might even work with BPTT.

**Questions:**

Sentence in lines 88-90: I don't follow. Please unpack.

Lines 38-39. This assertion is a bit strong. There have been some preliminary attempts to explain human RTs using image-computable models so it would be good to cite them.

**Limitations:**

As stated above, would be useful to give some examples of the challenges which may arise when applying this approach to other types of tasks.

---

> ### Author Rebuttal · Authors · 2023-08-10
>
> We thank the reviewer for their overall positive feedback and insightful comments.
>
> >**Does instantaneous uncertainty have to go to zero? What happens in the case of ambiguous stimuli?**
>
> We thank the reviewer for this observation and their suggestion to test the model on ambiguous stimuli.
>
> We take this opportunity to clarify that while our training procedure promotes stable attractor states in the model's *latent* representation, this does not directly correspond to a model reaching an uncertainty (obtained from the *readout*) value of *zero*. Additionally, given that we operate within a discrete, finite-time horizon paradigm, our metric does indeed remain bounded. In **Fig. R2**, we present one such example where a challenging stimulus results in an attractor state with non-zero uncertainty.
>
> We also curated a novel dataset consisting of "occluded" stimuli, a classic scenario of ambiguity (**Fig R1a**). We perform zero-shot transfer of our trained model to this novel test set and observe its uncertainty traces which do match the reviewer's intuition (**Fig. R1a**). A paired t-test reveals that our metric is significantly higher in the Occluded condition compared to the Control condition: $Cohen's\ d =  2.85,  t(37) = 17.54, p < .001$ (**Fig R1b**). We will add this analysis as a part of the discussion in the manuscript.
>
> >**Biological plausibility of the metric**
>
> Excellent point! We agree that metrics mined from an RL-trained model are likely to have analogous biological correspondences (as opposed to models trained via supervision). However, this is a hypothesis that needs empirical confirmation. We are excited to pursue this line of research in future work.
>
> >**Sentence in lines 88-90: I don't follow. Please unpack.**
>
> We apologize for the confusing wording. We will make the correction as follows:
>
> The elegant approach of [26] suffers from the primary drawback that it relies on the backpropagation through time (BPTT) algorithm. BPTT imposes high memory demands, limiting the number of timesteps a cRNN can perform and therefore condemning any derived measure to be coarse. Furthermore, vanilla BPTT forces the dynamics to be constant (i.e., the cRNN arrives at a solution at t = T for any input) making it non-stimulus-adaptive [8].
>
> >**Lines 38-39. This assertion is a bit strong. There have been some preliminary attempts to explain human RTs using image-computable models so it would be good to cite them.**
>
> Thanks for the comment. We will include the following references as prior art in the domain of explaining human RTs using image-computable models. We shall also highlight better reference [26] from the current main text.
>
> Mirzaei et al. (2013) Predicting the human reaction time based on natural image statistics in a rapid categorization task. Vision Research, 81(5), 36-44.
>
> Kumbhar et al. (2020) Anytime Prediction as a Model of Human Reaction Time. arXiv preprint arXiv:2011.12859.
>
> Duinkharjav et al. (2022) Image Features Influence Reaction Time: A Learned Probabilistic Perceptual Model for Saccade Latency. ACM Transactions on Graphics, 41(4), 1–15.

---

### Author Rebuttal · Authors · 2023-08-10

We thank the reviewers for their time in reading our manuscript and for their extensive feedback. In this general response, we address some common themes across the reviews. We provide detailed answers to specific reviewers' comments in subsequent responses. To go with this rebuttal, we also provide a PDF with additional figures labeled **Fig. R1-R5**, as well as an interactive version of our SU maps (URL supplied to the AC).

First, we sincerely thank the reviewers for their overwhelmingly positive feedback. All reviewers noted the importance and novelty of our work, as well as the thoroughness of our experiments.

One common critique across reviews pointed towards the need for a comparison across model architectures and alternative training frameworks.

We do believe that our framework is broadly applicable. And towards addressing questions raised by the reviewers, we perform and include several new analyses in this rebuttal. Specifically,

1. To demonstrate the generality of our metric, we trained a convolutional LSTM (convLSTM) model on the scene categorization task and show that our metric can work with alternative model architectures as well (**Fig. R3**) (More details in our response to **JuhY**)

1. We *implemented* and made a direct comparison to a model that explicitly has adaptive computational time (ACT) trained on our incremental grouping task (as suggested by **4B3C**). More details in our response to **4B3C**.

1. We probe the generalization benefits of models exhibiting temporal alignment with humans (RTs). Specifically, we demonstrate that poorly-aligned models have a propensity to fail at out-of-distribution generalization while well-aligned models are consistently good in their generalization performance. We refer to our response to **fhSL** for more details in this regard.

We hope you agree that our manuscript has improved through your feedback and that our findings will have a significant impact on computational cognitive neuroscience and machine learning.

---

### Decision · Program_Chairs · 2023-09-21

**Decision:**

Accept (spotlight)

**Comment:**

The paper proposes a recurrent vision model that accounts for human reaction times in visual decision making. All reviewers agree that it is a strong paper that makes a valuable contribution. One issue raised in the reviews and discussion was that the metric the paper proposes is more akin to uncertainty than reaction time, but this issue does not appear to undermine the overall value of the paper but rather provides some food for thought and future work.